# DEEP CLUSTERING WITH ASSOCIATIVE MEMORIES

## ABSTRACT

Deep clustering – joint representation learning and latent space clustering – is a well studied problem especially in computer vision and text processing under the deep learning framework. While the representation learning is generally differentiable, clustering is an inherently discrete optimization task, requiring various approximations and regularizations to fit in a standard differentiable pipeline. This leads to a somewhat disjointed representation learning and clustering. Recently, Associative Memories were utilized in the end-to-end differentiable `ClAM` clustering scheme (Saha et al., 2023). In this work, we show how Associative Memories enable a novel take on deep clustering, `DClAM`, simplifying the whole pipeline and tying together the representation learning and clustering more intricately. Our experiments showcase the advantage of `DClAM`, producing improved clustering quality regardless of the architecture choice (convolutional, residual or fully-connected) or data modality (images or text).

## 1 INTRODUCTION

The goal of clustering is to find coherent groups in a dataset. It is an important unsupervised learning task, and given the generality of the task, many different methods have been proposed for effective clustering (Xu & Tian, 2015; Zaki & Meira Jr, 2020). At a technical level, clustering critically relies on a notion of (pairwise) distance (or similarity) to distinguish pairs of data samples as being "similar" or "different", and the insights from clustering can be unintuitive or misleading without such a meaningful distance. When dealing with numerical data $S \subset \mathbb{R}^d$ with $d$ dimensions, metrics such as Euclidean distance are commonly used. Nevertheless, even with numerical data and an appropriate notion of distance, increasing data dimensionality (that is, increasing $d$) makes clustering computationally hard as well as conceptually difficult since the separation between similar pairs and dissimilar ones can start to vanish (Verleysen & François, 2005; Steinbach et al., 2004; Assent, 2012).

In various domains, both these problems manifest – first, the raw representation of samples can be extremely high dimensional (consider the number of pixels in an image, or the number of words in a vocabulary for a bag-of-words representation of documents); second, while we have an *ambient* representation, standard notions of vector distances (such as Euclidean) do not necessarily make sense – for example, Euclidean distance based on pixels can be large between an image and a slightly shifted version of it, which can be problematic if the content of an image is translation or rotation invariant.

One effective approach to handle these challenges is through *deep clustering* (Zhou et al., 2024), where the goal is to both learning a low dimensional *latent* space where standard distance metrics are meaningful, and to cluster or group the points at the same time. For the latent representations to be faithful to the original samples, deep clustering ensures that there is no significant information loss in the latent space, leading to the common use of autoencoders (AEs) (Rumelhart et al., 1985; Baldi, 2012; Bank et al., 2023) that learn latent representations (via an encoder) which can be used to reconstruct the original samples (via a decoder). The goal of deep clustering is to discover the cluster structure in the latent space while ensuring low reconstruction loss. This is a widely studied problem, especially in image datasets (Caron et al., 2018; Chang et al., 2017). While an autoencoder is usually differentiable, standard clustering schemes (such as $k$-means (MacQueen, 1967) or agglomerative (Johnson, 1967)) are inherently discrete methods, since *hard* clustering (where each sample is only assigned to a single cluster) is a discrete optimization problem. To incorporate it in a differentiable deep learning pipeline, clustering is often "softened" by allowing samples to be partially assigned to multiple clusters, although various "regularizations" push the soft assignments to

match hard assignments approximately (Xie et al., 2016; Guo et al., 2017a). The recent `ClAM` (Saha et al., 2023) algorithm handles the dichotomy between hard assignments and differentiability via the use of associative memories, yielding an end-to-end differentiable clustering approach. Nevertheless, `ClAM` works only in the ambient $d$-dimensional data space, and is not designed to learn effective lower dimensional latent representations, which poses challenges when clustering high-dimensional data.

As noted above, deep clustering tackles the joint objective of learning a good latent representation where the points also cluster well. Whereas minimizing reconstruction loss is a prerequisite for deep representation learning, one option for clustering in latent space is to first *pretrain* an autoencoder to minimize the reconstruction loss, and then to freeze this latent space. Next, one can apply some clustering scheme to group the points in that (frozen) space. Many AE based existing deep learning methods adopt this scheme by either freezing both the encoder and decoder, or freezing only the decoder (Xie et al., 2016; Guo et al., 2017b; 2021; Chazan et al., 2019; Huang et al., 2023).

In this paper, we consider a new approach that fine-tunes the autoencoder (i.e., both the encoder and decoder) so that it modifies the latent space along with the task of finding clusters, while also ensuring end-to-end differentiabilty via the use of associative memories (significantly extending (Saha et al., 2023)). Our key insight and contribution is that we seamlessly combine the clustering and reconstruction loss objectives into one expression that tackles the task of **clustering-guided latent representations**, whereas previous deep clustering methods considered these separately. Our work makes the following contributions:

- We propose **DClAM**, which uses associative memories to formulate a novel joint loss function that simultaneously learns effective representations and clusters in the latent space, resulting in our simplified deep clustering formulation.
- We conduct a thorough evaluation on image and text datasets, demonstrating that **DClAM** significantly improves the clustering quality over both traditional (in ambient space) and deep clustering (in latent space) baselines.
- We show that **DClAM** retains superior representation quality as measured by the reconstruction loss; it is also agnostic to the encoder/decoder architecture choice.

## 2 RELATED WORK

Clustering is a long-studied and well-reviewed problem in computer science, with various formulations and several applications (Kaufman & Rousseeuw, 2009; Zaki & Meira Jr, 2020). Given the success of deep learning, deep clustering has also attracted attention over the past decade (Ren et al., 2024; Aljalbout et al., 2018; Zhou et al., 2024). Inspired by t-SNE (Van der Maaten & Hinton, 2008), Xie et al. (2016) introduced DEC, enhancing clustering and feature representation by minimizing the Kullback-Leibler Divergence (KLD) to an auxiliary target distribution. However, a drawback is abandoning the decoder layer after pre-training, impacting the embedded space and clustering performance. Guo et al. (2017a) showed that keeping the decoder layer improves clustering (IDEC), and Guo et al. (2017b) proposed DCEC using convolutional autoencoders (CAE). Chazan et al. (2019) proposed DAMIC, a mixture of autoencoders for clustering, determined by minimizing the reconstruction loss without needing a regularization term. However, they leverage multiple AEs in their model, while we focus on schemes using a single AE. Huang et al. (2023) introduced an innovative embedded autoencoder architecture by incorporating it into both the encoding and decoding units of the outer autoencoder. Guo et al. (2021) proposed DEKM which works on the embedding space (after pretraining) and transforms it to a new cluster-friendly space using an orthonormal transformation matrix. However, discarding the decoder after pretraining for both of these methods may lead to the distortion of the embedded space, consequently hurting clustering performance. In addressing the automatic inference of the number of clusters in a dataset, Ronen et al. (2022) introduced DeepDPM. They proposed a novel loss inspired by EM in the Bayesian Gaussian Mixture Model framework, facilitating a new amortized inference in mixture models. It is worth noting that DeepDPM diverges from the typical encoder-decoder architecture, opting instead for a multilayer perceptron model.

While many deep clustering methods utilize KLD as a clustering objective, it falls short in preserving the global data structure (which implies that only within-cluster distances are prioritized, leaving uncertainties regarding between-cluster similarities), leading Oskouei et al. (2023) (EDCWRN) to advocate for cross-entropy over KLD. They incorporate feature weighting to emphasize essential

features for clustering and employ a neighborhood technique to encourage similar representations for samples within the same cluster. Addressing another challenge with KLD regarding the presence of hard, misclassified samples, Cai et al. (2022) introduced focal loss to enhance label assignment in deep clustering methods and improved the representation learning module with a contractive penalty term, capturing more discriminative representations. However, it could lead to unintentional bias in the optimization focus between the representation learning and clustering modules. Dang et al. (2021) introduce a novel deep clustering framework (NNM) based on a two-level nearest neighbors matching approach. Distinguishing itself from prior methods (Van Gansbeke et al., 2020), NNM incorporates matching at both local and global levels, resulting in a notable enhancement in clustering performance. It also leverages SimCLR (Chen et al., 2020) to pretrain a representation learning model using the state-of-the-art contrastive learning loss. In our work, we rethink the deep clustering problem at a architecture agnostic level by leveraging the capabilities of associative memories. Thus, various architectural and pretraining advancements would also benefit our proposed scheme.

Recently, Saha et al. (2023) introduced `ClAM`, an end-to-end differentiable clustering approach, utilizing Dense Associative Memories (AMs) for clustering. AMs adeptly store multidimensional vectors as fixed point attractor states in a recurrent dynamical system. AMs form associations between the initial state and a final state (memory), creating disjoint basins of attractions which are crucial for clustering. A prominent example of AM is the classical Hopfield Network (Hopfield, 1982). It exhibits limited memory capacity, approximately storing only $\approx 0.14d$ arbitrary memories in a $d$ dimensional data domain (McEliece et al., 1987; Amit et al., 1985). Subsequently, Krotov & Hopfield (2016) proposed Dense Associative Memory (Dense AM) or Modern Hopfield Network introducing rapidly growing non-linearities (activation functions) into the system. This innovation allows for a denser arrangement of memories and achieves super-linear (in $d$) memory capacity (Demircigil et al., 2017; Lucibello & Mézard, 2023). With softmax activation, Dense AMs are closely related to the attention mechanism used in transformers (Ramsauer et al., 2020; Krotov & Hopfield, 2021; Hoover et al., 2024). Further, Schaeffer et al. (2023) demonstrates that the energy function of `ClAM`'s AM network is equal to a scaled negative log-likelihood of a Gaussian mixture model. In our work, we study the joint task of learning effective latent representations and clustering in the latent space. We continuously refine both the encoder and decoder networks and at the same time integrate the AM learning dynamics to cluster the points into $k$ groups. This bears semblance to vector-quantized variational AEs (van den Oord et al., 2017), where the task is to learn a discrete vector code for each point. Nevertheless, this assignment is non-differentiable, requiring gradient approximation, and there is no clustering objective considered. Also related is the task of deep metric learning (Kaya & Bilge, 2019), where the task is to learn a distance function between samples in latent space. Nevertheless, this requires the use of labeled data for full or weak supervision. As such, the coupling of deep clustering with AMs as done in `DClAM` for cluster-guided latent space learning has not been considered in the literature before.

# 3 PRELIMINARIES

We denote an input set as $S \subset \mathbb{R}^d$ in the ambient space, with an input $x \in S$, and $[\![n]\!]$ a $n$-length index set $\{1, \ldots, n\}$.

## 3.1 DEEP CLUSTERING BASICS

Deep clustering is an unsupervised task, where we have to learn (usually lower dimensional) representations such that (i) no (critical) information is lost in the latent lower dimensional representations, and (ii) the data in the latent space forms well-separated clusters. To ensure that no information is lost in the latent space, we learn an encoder $\mathbf{e} : \mathbb{R}^d \to \mathbb{R}^m$ ($m < d$) that maps the input $x \in \mathbb{R}^d$ to a latent space (that is, $\mathbf{e}(x) \in \mathbb{R}^m$), along with a decoder $\mathbf{d} : \mathbb{R}^m \to \mathbb{R}^d$ that maps the latent representation back to the original ambient space. Encoder $\mathbf{e}$ and decoder $\mathbf{d}$ together give us an autoencoder, and the loss of information is often measured as the *reconstruction loss*:

$$\mathcal{L}_r(\mathbf{e}, \mathbf{d}) = \sum_{x \in S} \ell_r(x, \mathbf{e}, \mathbf{d}) = \sum_{x \in S} \|x - \mathbf{d}(\mathbf{e}(x))\|^2. \tag{1}$$

This loss term does not account for the cluster structure in the latent space. For that purpose, we consider $k$ cluster centers $\boldsymbol{\rho} = \{\rho_1, \ldots, \rho_k\} \subset \mathbb{R}^m$ in the latent space, so that the corresponding

*clustering loss* is given by:

$$\mathcal{L}_c(\mathbf{e}, \boldsymbol{\rho}) = \sum_{x \in S} \ell_c(x, \mathbf{e}, \boldsymbol{\rho}) = \sum_{x \in S} \min_{i \in [\![k]\!]} \|\mathbf{e}(x) - \rho_i\|^2, \tag{2}$$

which measures how close a sample is to its closest cluster center in the latent space with the $\min_{i \in [\![k]\!]}$ performed on a per-sample basis to denote the discrete assignment. A small value of $\mathcal{L}_c(\mathbf{e}, \boldsymbol{\rho})$ implies that all points in the latent space are close to their respective cluster centers.

Unsupervised deep clustering is often considered in the following form (Guo et al., 2017a;b; Cai et al., 2022)

$$\min_{\mathbf{e}, \mathbf{d}, \boldsymbol{\rho}} \mathcal{L}_r(\mathbf{e}, \mathbf{d}) + \gamma \mathcal{L}_c(\mathbf{e}, \boldsymbol{\rho}) \tag{3}$$

where $\gamma \geq 0$ is a hyperparameter that balances the clustering loss $\mathcal{L}_c$ and the reconstruction loss $\mathcal{L}_r$. This $\gamma$ plays a critical role in balancing the two terms in Eq. (3).

While this hyperparamter $\gamma$ can be handled via hyperparameter optimization, there is an inherent challenge in the above objective — the terms $\mathcal{L}_c$ and $\mathcal{L}_r$ are not inherently comparable. The per-sample clustering loss $\ell_c(x, \mathbf{e}, \boldsymbol{\rho})$ is a loss computed between entities in the latent space $\mathbb{R}^m$, while per-sample reconstruction loss $\ell_r(x, \mathbf{e}, \mathbf{d})$ is a loss between items in the ambient space $\mathbb{R}^d$. Thus, the scale of these two terms can be very different, making it hard to select a good value for $\gamma$.

Usual implementations of deep clustering (Guo et al., 2017a;b; Oskouei et al., 2023) adopt the following strategy: (i) First, an autoencoder (that is, $\mathbf{e}$ and $\mathbf{d}$) is "pretrained" with the data to achieve low reconstruction error (that is, low $\mathcal{L}_r$ by setting $\gamma = 0$ in Eq. (3)), and (ii) second, $\gamma$ is set to a positive value in Eq. (3), and the clustering loss $\mathcal{L}_c$ is minimized by learning the cluster centers $\boldsymbol{\rho}$, and "fine-tuning" the encoder $\mathbf{e}$, while the reconstruction loss $\mathcal{L}_r$ stays low by changing the decoder $\mathbf{d}$ accordingly *if the balancing hyperparameter $\gamma$ is set appropriately, which can be a challenge.*

### 3.2 DENSE ASSOCIATIVE MEMORIES AND CLUSTERING

Given $k$ memories $\{\boldsymbol{\rho}_1, \ldots, \boldsymbol{\rho}_k\}$, $\boldsymbol{\rho}_i \in \mathbb{R}^d$, and a point or particle $v \in \mathbb{R}^d$, **ClAM** (Saha et al, 2023) defines the energy function for $v$ as follows:

$$E(v) = -\frac{1}{\beta} \log \left( \sum_{i \in [\![k]\!]} \exp(-\beta \|\boldsymbol{\rho}_i - v\|^2) \right) \tag{4}$$

with the scalar $\beta > 0$ playing the role of inverse "temperature". As $\beta$ increases, the $\exp(\cdot)$ function emphasizes the leading term, suppressing the others. In **ClAM**, the attractor dynamics are driven by gradient descent on the energy landscape. This controls the movement of $v$ over time through $dv/dt$, ensuring a decrease in energy:

$$\tau \frac{dv}{dt} = -\frac{1}{2} \nabla_v E = \sum_{i \in [\![k]\!]} (\boldsymbol{\rho}_i - v) \, \mathsf{softmax}(-\beta \|\boldsymbol{\rho}_i - v\|^2) \tag{5}$$

Here, $\tau > 0$ is a characteristic time constant that determines how quickly the particle will move on the energy landscape, and $\mathsf{softmax}(\cdot)$ represents the softmax function applied to the scaled distances to the memories, given as

$$\mathsf{softmax}(-\beta \|\boldsymbol{\rho}_i - v\|^2) = \frac{\exp(-\beta \|\boldsymbol{\rho}_i - v\|_2^2)}{\sum_{j \in [k]} \exp(-\beta \|\boldsymbol{\rho}_j - v\|_2^2)} \tag{6}$$

Given the state vector $v^t$ as step $t$, the energy-based AM update for step $t + 1$ is given via gradient descent on the energy:

$$v^{t+1} = A_{\boldsymbol{\rho}}(v^t) = v^t + \tau \left. \frac{dv}{dt} \right|_{v=v^t} \tag{7}$$

Here we denote the AM update via the operator $A_{\boldsymbol{\rho}}(v)$ defined above, and we use the notation $A_{\boldsymbol{\rho}}^T(v)$ to denote $A_{\boldsymbol{\rho}}(A_{\boldsymbol{\rho}}(\cdots A_{\boldsymbol{\rho}}(v)))$, where the operator $A_{\boldsymbol{\rho}}$ is applied to $v$ recursively for $T$ steps. The attractor dynamics ensure that every memory $\rho_i, i \in [\![k]\!]$, forms a "basin of attraction", and with enough recursions $T$, any particle will usually converge to exactly one of these memories $\rho_i$, which

thus act as cluster centers. The differentiability of the recursive dynamics is what makes **ClAM** an end-to-end differentiable clustering scheme, with the memories learned via standard backpropagation. However, it is important to note that **ClAM** is not a deep clustering method, since there is no representation learning involved. Of course, one can apply **ClAM** in latent space learned by a pretrained autoencoder, but we show that this is not an effective strategy. Instead, we next propose our novel **DClAM** approach that is inherently a deep clustering method that jointly clusters and learns effective latent representations.

## 4   **DClAM**: Deep Clustering with Associative Memories

Existing deep clustering methods need to solve Eq. (3) explicitly, which involves the critical $\gamma$ hyperparameter to appropriately balance the clustering and reconstruction losses. Here, we will show how AM enables the removal of the critical $\gamma$ hyperparameter in the deep clustering objective (Eq. (3)), while still maintaining the intent of Eq. (3) to balance the clustering loss and the reconstruction loss.

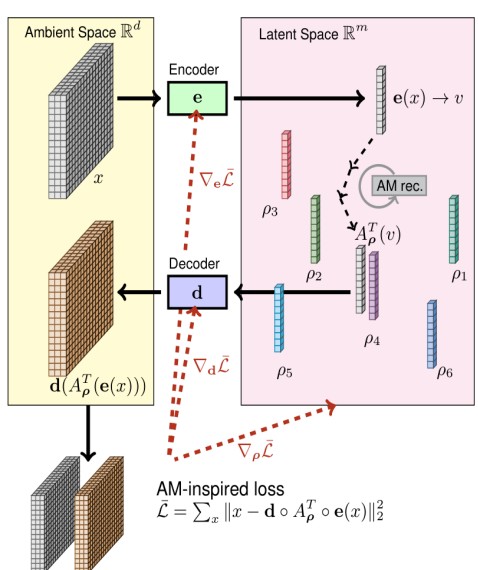

Figure 1: **DClAM**: AM-enabled deep clustering. The solid arrows $\longrightarrow$ denote the forward-pass to compute the single loss term in Eq. (8). The dashed arrows $-\rightarrow$ denote the backward pass showing the single loss driving all updates.

---

**Algorithm 1: DClAM Algorithm**

1   **Train**$(S, k, N, T, \epsilon_{\mathbf{e}}, \epsilon_{\mathbf{d}}, \epsilon_{\boldsymbol{\rho}}, \gamma)$
2    Pretrain $(\mathbf{e}, \mathbf{d})$ as autoencoder, minimizing $\mathcal{L}_r(\mathbf{e}, \mathbf{d})$
3    $\boldsymbol{\rho} \leftarrow \{\mathbf{e}(x), x \in M\}$, $M$ are random $k$ samples from S
4    **for** *epoch* $n = 1, \ldots, N$ **do**
5      **for** *batch* $B \in S$ **do**
6        Batch loss $\bar{\mathcal{L}}_B \leftarrow 0$
7        **for** *example* $x \in B$ **do**
8          $v \leftarrow \mathbf{e}(x)$ // encode
9          $\bar{v} \leftarrow A_{\boldsymbol{\rho}}^T(v)$ // AM steps
10         $\bar{\ell} \leftarrow \|x - \mathbf{d}(\bar{v})\|^2$ // loss
11         $\bar{\mathcal{L}}_B \leftarrow \bar{\mathcal{L}}_B + \ell$
12        $\rho_i \leftarrow \rho_i - \epsilon_{\boldsymbol{\rho}} \nabla_{\rho_i} \mathcal{L}_B \; \forall i \in [\![k]\!]$
13        $\mathbf{e} \leftarrow \mathbf{e} - \epsilon_{\mathbf{e}} \nabla_{\mathbf{e}} \mathcal{L}_B$
14        $\mathbf{d} \leftarrow \mathbf{d} - \epsilon_{\mathbf{d}} \nabla_{\mathbf{d}} \mathcal{L}_B$
15    **return** $\mathbf{e}, \mathbf{d}, \boldsymbol{\rho}$

16   **Infer**$(S, \mathbf{e}, \mathbf{d}, \boldsymbol{\rho})$
17    Cluster assignments $C \leftarrow \emptyset$
18    **for** $x \in S$ **do**
19      $\bar{v} \leftarrow A_{\boldsymbol{\rho}}^T(\mathbf{e}(x)) \; C \leftarrow$   $C \cup \left\{ \arg\min_{i \in [\![k]\!]} \|\rho_i - \bar{v}\|^2 \right\}$
20    **return** *Per-point cluster assignments* $C$

---

Recall from Eq. (7) that $A_{\boldsymbol{\rho}}(v^t)$ denotes the AM dynamics step, given as $v^{t+1} = A_{\boldsymbol{\rho}}(v^t) = v^t + \tau \frac{dv^t}{dt}$, where $v^{t+1}$ is the new state vector obtained by updating $v^t$, and recall that $A_{\boldsymbol{\rho}}^T(v)$ denotes the dynamics for $T$ steps. Consider the pipeline depicted in Fig. 1: The input $x$ is mapped into the latent space as $\mathbf{e}(x)$ by the encoder $\mathbf{e}$, and then the attractor dynamics operator $A_{\boldsymbol{\rho}} : \mathbb{R}^m \to \mathbb{R}^m$ based on the current centers $\boldsymbol{\rho} = \{\rho_1, \ldots, \rho_k\}$ is applied to $\mathbf{e}(x)$ for $T$ recursions, resulting in $A_{\boldsymbol{\rho}}^T(\mathbf{e}(x))$ (which will be close to one of the cluster centers). Then this representation (effectively of a cluster center) is passed through the decoder $\mathbf{d}$ to get $\mathbf{d}(A_{\boldsymbol{\rho}}^T(\mathbf{e}(x))) \in \mathbb{R}^d$ in the ambient space. We can then optimize for the following loss:

$$\min_{\mathbf{e}, \mathbf{d}, \boldsymbol{\rho}} \bar{\mathcal{L}}(\mathbf{e}, \mathbf{d}, \boldsymbol{\rho}) = \sum_{x \in S} \bar{\ell}(x, \mathbf{e}, \mathbf{d}, \boldsymbol{\rho}) = \sum_{x \in S} \left\| x - \mathbf{d} \left( A_{\boldsymbol{\rho}}^T \left( \mathbf{e}(x) \right) \right) \right\|^2 \qquad (8)$$

Here AM becomes the intricate part of the encoder that transforms the embedding space (obtained by the encoder) into a clustering-friendly new space to find clusters (as opposed to the existing deep clustering schemes that use different additional loss functions, e.g., clustering loss in Eq. (3) and/or regularizations to get a similar effect). This AM enabled *novel deep clustering loss* $\bar{\mathcal{L}}$ is a single

term that elegantly combines all the parameters in the deep learning pipeline – for the encoder $\mathbf{e}$, the cluster centers $\boldsymbol{\rho}$ and the decoder $\mathbf{d}$.

To see the relationship between the novel `DC1AM` loss compared to the traditional deep clustering loss in Eq. (8), let us assume that the decoder $\mathbf{d} : \mathbb{R}^m \to \mathbb{R}^d$ is $C_{\mathbf{d}}$-Lipschitz continuous. Then, considering the per-sample loss $\bar{\ell}$ in Eq. (8), and applying the triangle inequality and the AM–GM inequality, we can show that

$$
\begin{aligned}
\bar{\ell}(x, \mathbf{e}, \mathbf{d}, \boldsymbol{\rho}) &= \|x - \mathbf{d}(A_{\boldsymbol{\rho}}^T(\mathbf{e}(x)))\|^2 \\
&\leq 2 \left( \|x - \mathbf{d}(\mathbf{e}(x))\|^2 + \|\mathbf{d}(\mathbf{e}(x)) - \mathbf{d}(A_{\boldsymbol{\rho}}^T(\mathbf{e}(x)))\|^2 \right) \\
&\leq 2 \left( \|x - \mathbf{d}(\mathbf{e}(x))\|^2 + C_{\mathbf{d}}^2 \|\mathbf{e}(x) - A_{\boldsymbol{\rho}}^T(\mathbf{e}(x))\|^2 \right) \\
&= 2\ell_r(x, \mathbf{e}, \mathbf{d}) + 2C_{\mathbf{d}}^2 \ell_c(x, \mathbf{e}, \boldsymbol{\rho}),
\end{aligned}
\tag{9}
$$

where the last inequality uses the Lipschitz continuity, and the last equality comes from the definition of the clustering loss in the latent space with the AM dynamics operator. Summing the above inequalities over $x \in S$ gives us

$$
\mathcal{L}_r \ \leq \ \bar{\mathcal{L}} \ \leq \ \gamma_1 \mathcal{L}_r + \gamma_2 \mathcal{L}_c
\tag{10}
$$

where the upperbound of $\bar{\mathcal{L}}$ is (a scaled version of) the standard deep clustering objective of the weighted combination of the reconstruction loss $\mathcal{L}_r$ and the clustering loss $\mathcal{L}_c$ in Eq. (3).

Alg. 1 shows the pseudo-code for `DC1AM`. It first pretrains encoder $\mathbf{e}$ and decoder $\mathbf{d}$, and starts from $k$ random prototypes $\boldsymbol{\rho}$. The cluster assignment is done with $T$ recursion of the AM attractor dynamics operator $A_{\boldsymbol{\rho}}$ parameterized with the centers $\boldsymbol{\rho} = \{\rho_i, i \in [\![k]\!]\}$. The per-sample loss $\bar{\ell}$ of `DC1AM` (line 10) is added to the batch loss. We optimize for $N$ epochs via gradient descent, with learning rates $\{\epsilon_{\mathbf{e}}, \epsilon_{\mathbf{d}}, \epsilon_{\boldsymbol{\rho}}\}$ for $\mathbf{e}, \mathbf{d}, \boldsymbol{\rho}$ respectively. Upon solving Eq. (8), we obtain a trained encoder and decoder, and memories in the latent space, and we can utilize them to obtain the final partition the data (see the **Infer** subroutine in Alg. 1).

Our `DC1AM` deep clustering provides various advantages: (i) First, it does not involve any balancing hyperparameter $\gamma$ since the loss involves all parameters in a single term in the per-sample loss $\bar{\ell}(x, \mathbf{e}, \mathbf{d}, \boldsymbol{\rho})$. (ii) Second, the updates for all the parameters in the pipeline are more explicitly tied together with the $\mathbf{d} \circ A_{\boldsymbol{\rho}}^T \circ \mathbf{e}$ composition in the $\mathbf{d}(A_{\boldsymbol{\rho}}^T(\mathbf{e}(x)))$ term. *This ties the representation learning and clustering objectives more intricately.* (iii) Third, it continues to have all the advantages of traditional deep clustering, being end-to-end differentiable since all operators in the above composition are differentiable, and performing a discrete cluster center assignment with $T$ recursions of the attractor dynamics operator $A_{\boldsymbol{\rho}}$. (iv) Forth, this deep clustering is completely architecture agnostic – we can select a problem dependent encoder and decoder (for example, convolutional or residual networks for images or fully-connected feed-forward networks for text or tabular data). (v) Fifth, it does not involve any additional entropy regularization based hyperparameters as with existing deep clustering algorithms.

On a less technical level, Fig. 1 clearly highlights how the overall information flow in the deep clustering pipeline is simplified. *The AM plays a critical role in this pipeline with the ability to obtain the actual closest center $A_{\boldsymbol{\rho}}^T(\mathbf{e}(x))$ in a differentiable manner.* It is also worth emphasizing that there are fundamental difference between how `DC1AM` uses AMs versus `C1AM`. In the latter, AM is utilized to act as a differentiable argmin solver for the $k$-means objective whereas in `DC1AM`, which involves representation learning, AM recursion actually has a more elaborate effect. The AM augmented encoder ($A_{\boldsymbol{\rho}}^T \circ \mathbf{e}$) explicitly creates basins of attraction in the latent space, and moves/pushes the latent representations of the points into these basins, thereby explicitly inducing a clustered data distribution in the latent space. While the encoder is moving points into basins of attraction, the `DC1AM` loss tries to minimize the information loss in the latent representations by having the decoder reconstruct these relocated latent representations.

## 5 EMPIRICAL EVALUATION

**Dataset details.** To evaluate `DC1AM`, we conducted our experiments on eight standard benchmark datasets, including USPS[1] (Hull, 1994), Fashion-MNIST[2] (Xiao et al., 2017), CIFAR-10 and CIFAR-

---

[1] https://www.kaggle.com/datasets/bistaumanga/usps-dataset
[2] https://github.com/zalandoresearch/fashion-mnist

$100^3$ (Krizhevsky, 2009), STL-$10^4$ (Coates et al., 2011), Caltech_birds2010[5] (Welinder et al., 2010), 20-NG from sklearn[6] and Reuters-10k from TensorFlow datasets[7]. The later two are text datasets, whereas the others are image datasets. For both text datasets, we calculate TFIDF (Sammut & Webb, 2010) features based on the 2000 most frequent words, following a similar approach as Oskouei et al. (2023). However, we consider the original number of categories as the true number of clusters, which is 46 for Reuters-10k and 20 for 20-NG. For Caltech_birds2010, as there are images of various shapes, we resize all images to (128, 128, 3) for uniformity and ease of implementation. Table 1 provides the statistics for the datasets used in our experiments.

Table 1: Descriptions of various benchmark datasets, used in our experiments.

| Dataset | Short name | # Points | Shape | # Classes | # Type |
|---|---|---|---|---|---|
| Fashion MNIST | FM | 60000 | (28, 28, 1) | 10 | Image |
| CIFAR-10 | C-10 | 50000 | (32, 32, 3) | 10 | Image |
| CIFAR-100 | C-100 | 50000 | (32, 32, 3) | 100 | Image |
| USPS | USPS | 2007 | (16, 16, 1) | 10 | Image |
| STL-10 | STL | 5000 | (96, 96, 3) | 10 | Image |
| Caltech_birds2010 | CBird | 3000 | (128, 128, 3) | 200 | Image |
| Reuters-10k | R-10k | 11228 | 2000 | 46 | Text |
| 20-NG | 20NG | 18846 | 2000 | 20 | Text |

**Baseline methods.** We conduct a comparative analysis of **DClAM** against established clustering methods like $k$-means (Lloyd, 1982), agglomerative clustering (or Agglo.) (Müllner, 2011) and **ClAM** (Saha et al., 2023) in the ambient space, and with DCEC (Guo et al., 2017b), DEKM (Guo et al., 2021) and EDCWRN (or EDC) Oskouei et al. (2023) deep clustering methods in the latent space. We evaluate $k$-means, agglomerative clustering, and **ClAM** in the ambient space (denoted as NAE for No AE) and in the latent space obtained through a pretrained Convolutional Autoencoder (CAE) as used in DCEC (Guo et al., 2017b). For DCEC amd DEKM, we consider a ResNet-based AE (RAE) (Wickramasinghe et al., 2021) along with their original CAE. For **DClAM**, we extend our exploration to include not only the CAE and RAE architectures but also EDCWRN-based (Oskouei et al., 2023) Autoencoder (EAE) (originally proposed by Guo et al. (2017a)) to analyze its impact on the algorithm. We also compare **DClAM** with state-of-the-art SimCLR (Chen et al., 2020) based (contrastive learning) SCAN (Van Gansbeke et al., 2020) and NNM (Dang et al., 2021) deep clustering schemes. Detailed parameter setting of the networks are in Appendix A.2, while implementation details are in Appendix A.3.

**Evaluation of deep clustering.** A common metric to evaluate and benchmark deep clustering algorithms is by computing the overlap between the obtained clusters in the latent space (thus, partitions) and a semantic partitioning of the data (for example, using some ground-truth labels of the data that were not used for solving Eq. (3)) with metrics such as the Normalized Mutual Information (NMI) (Vinh et al., 2009). While this is a fair metric to compare methods on, *it is critical to ensure that NMI (or any other label-dependent metric) is not utilized for hyperparameter selection* since that is leaking supervision into the unsupervised task of deep clustering, making the overall process a supervised learning pipeline. Unfortunately, for many of reported results, it is not clear how hyperparameters are selected without being influenced at NMI (since they simply report results with the highest NMI). Even for the sole purpose of evaluation, NMI like metrics might only tell us how the learned clusters in the latent space match some semantic partitioning (often manual) of the data, and do not provide any information regarding the reconstruction quality (and thus the information loss in the latent space). Thus, it is easily possible to have high NMI with poor reconstruction loss, which may not align with the primary goals of deep clustering. Existing literature typically report NMI without explicitly discussing reconstruction loss.

---

[3] https://www.cs.toronto.edu/~kriz/cifar.html
[4] https://cs.stanford.edu/~acoates/stl10/
[5] https://www.tensorflow.org/datasets/catalog/caltech_birds2010
[6] https://scikit-learn.org/0.19/datasets/twenty_newsgroups.html
[7] https://www.tensorflow.org/api_docs/python/tf/keras/datasets/reuters/load_data

We believe that the hyperparameters should be selected based on unsupervised metrics – metrics that do not utilize any ground-truth label information to evaluate clustering quality – given the unsupervised nature of the deep clustering problem. Thus, we consider the strategy of optimizing for the Silhouette Coefficient (SC) (Rousseeuw, 1987) while keeping the reconstruction loss (RL) below some user-defined threshold. See Appendix A.1 for details on other metrics used.

Table 2: Per-method best SC across all architectures (while RRL is within 10% of the respective pretrained AE loss), comparing **DClAM** to baselines. Best for each dataset is in bold. See text for further details. *Higher SC is better, but lower RRL is better.* The top set of rows are vision datasets, and the bottom set are text datasets. A '-' indicates not applicable (NA); e.g., DCEC, DEKM, SCAN, NNM work only on image datasets. Further, we report SCAN and NNM results only on C-10, C-100 and STL, since these are the only datasets for which pretrained contrastive encoders are available. $x^{\blacktriangledown}$ indicates negative RRL which means the RL of the method is x% less than the pretrained AE loss.

| Dataset | SC | | | | | | | | | RRL | | | |
|---|---|---|---|---|---|---|---|---|---|---|---|---|---|
| | $k$-means | Agglo. | **ClAM** | DCEC | DEKM | EDC | SCAN | NNM | **DClAM** | DCEC | DEKM | EDC | **DClAM** |
| FM | 0.257 | 0.201 | 0.279 | 0.923 | 0.260 | 0.483 | - | - | **0.970** | 9.8 | 13.9$^{\blacktriangledown}$ | 10 | 1.6$^{\blacktriangledown}$ |
| C-10 | 0.084 | 0.372 | 0.208 | 0.787 | 0.116 | 0.511 | 0.541 | 0.587 | **0.863** | 9.6 | 8.6 | 10 | 19.5$^{\blacktriangledown}$ |
| C-100 | 0.015 | 0.149 | 0.053 | 0.470 | -0.007 | 0.311 | 0.321 | 0.358 | **0.598** | 7.5 | 34.3$^{\blacktriangledown}$ | 10 | 1.4$^{\blacktriangledown}$ |
| USPS | 0.195 | 0.158 | 0.194 | **0.935** | 0.217 | 0.461 | - | - | 0.891 | 5.3$^{\blacktriangledown}$ | 4.3 | 0.0 | 8.7 |
| STL | 0.079 | 0.270 | 0.108 | 0.259 | 0.082 | 0.411 | 0.552 | 0.540 | 0.891 | 9.2 | 0.6 | 4.9$^{\blacktriangledown}$ | 10 |
| CBird | -0.019 | 0.094 | -0.026 | 0.311 | -0.032 | 0.171 | - | - | **0.448** | 10 | 0.0 | 10 | 9.1 |
| R-10k | -0.010 | 0.114 | -0.002 | - | - | 0.023 | - | - | **0.564** | - | - | 10 | **10** |
| 20NG | -0.021 | 0.114 | -0.008 | - | - | 0.101 | - | - | **0.197** | - | - | 10 | **10** |

Table 3: Per-method best RRL across all architectures (while SC is within 10% of the best SC of the method) comparing **DClAM** to baselines. Best for each dataset is in bold. See text for further details. *Higher SC is better, but lower RRL is better.* $x^{\blacktriangledown}$ indicates negative RRL which means the RL of the method is x% less than the pretrained AE loss.

| Dataset | SC | | | | | | | | | RRL | | | |
|---|---|---|---|---|---|---|---|---|---|---|---|---|---|
| | $k$-means | Agglo. | **ClAM** | DCEC | DEKM | EDC | SCAN | NNM | **DClAM** | DCEC | DEKM | EDC | **DClAM** |
| FM | 0.257 | 0.201 | 0.279 | 0.898 | 0.785 | 0.521 | - | - | **0.922** | 9.8$^{\blacktriangledown}$ | 321 | 143 | 42.2 |
| C-10 | 0.084 | 0.372 | 0.208 | 0.786 | 0.622 | 0.541 | 0.541 | 0.587 | **0.809** | 0.9$^{\blacktriangledown}$ | 180 | 74.3 | 20.4$^{\blacktriangledown}$ |
| C-100 | 0.015 | 0.149 | 0.053 | 0.572 | 0.047 | 0.337 | 0.321 | 0.358 | **0.921** | 18.6 | 870 | 33.3 | 27.5 |
| USPS | 0.195 | 0.158 | 0.194 | **0.929** | 0.843 | 0.491 | - | - | 0.914 | 26.3$^{\blacktriangledown}$ | 2326 | 40 | 8.7 |
| STL | 0.079 | 0.270 | 0.108 | 0.812 | 0.804 | 0.431 | 0.552 | 0.540 | **0.923** | 79.2 | 234 | 155 | 27.7 |
| CBird | -0.019 | 0.094 | -0.026 | 0.282 | 0.018 | 0.188 | - | - | **0.413** | 286 | 1036 | 102 | 1.8 |
| R-10k | -0.010 | 0.114 | -0.002 | - | - | 0.035 | - | - | **0.673** | - | - | **60** | 120 |
| 20NG | -0.021 | 0.114 | -0.008 | - | - | 0.099 | - | - | **0.287** | - | - | 25$^{\blacktriangledown}$ | 50 |

**Q1: How does DClAM compare against baselines?** We present the best Silhouette Coefficient or SC achieved (while constraining the reconstruction loss or RL to be within 10% of the pretrained AE loss) by **DClAM**, and the baselines for all 8 datasets in Table 2. As it is hard to compare the raw RL numbers if the base AE is different for different methods, we define the relative RL (RRL) metric as follows:

$$RRL = \frac{RL - RL_p}{RL_p} \tag{11}$$

where $RL_p$ is the pretrained/base RL. We report the best SC per method with $RRL <= 10\%$. From Table 2, we see across both image and text datasets, **DClAM** consistently outperforms traditional and deep clustering baselines in terms of SC while keeping RRL relatively low. To provide a comprehensive view alongside SC, we also present the best RRL (while constraining the SC to be within 10% of the best/peak SC of the method) in Table 3 and visualize both SC and RL in Fig. 2 for all of the six image datasets. These results demonstrate that **DClAM** excels not only in achieving the best SC but also in minimizing RL compared to the baselines. Note that SCAN and NNM do not have a reconstruction loss term as they work on the pre-trained (pretext) model by SimCLR (Chen et al., 2020) and utilize only the encoder (discarding the decoder) for clustering purpose. For additional insights, in Appendix B.1, we present the best SC (while keeping RL within 10% of the pretrained AE loss) and its corresponding NMI, RL, and cluster sizes and balance obtained by all schemes in Table 8. Simultaneously, Table 9 displays the best RL (while keeping SC within 10% of the best SC of the method) and its associated SC, NMI, and cluster sizes. We also present Table 10 which

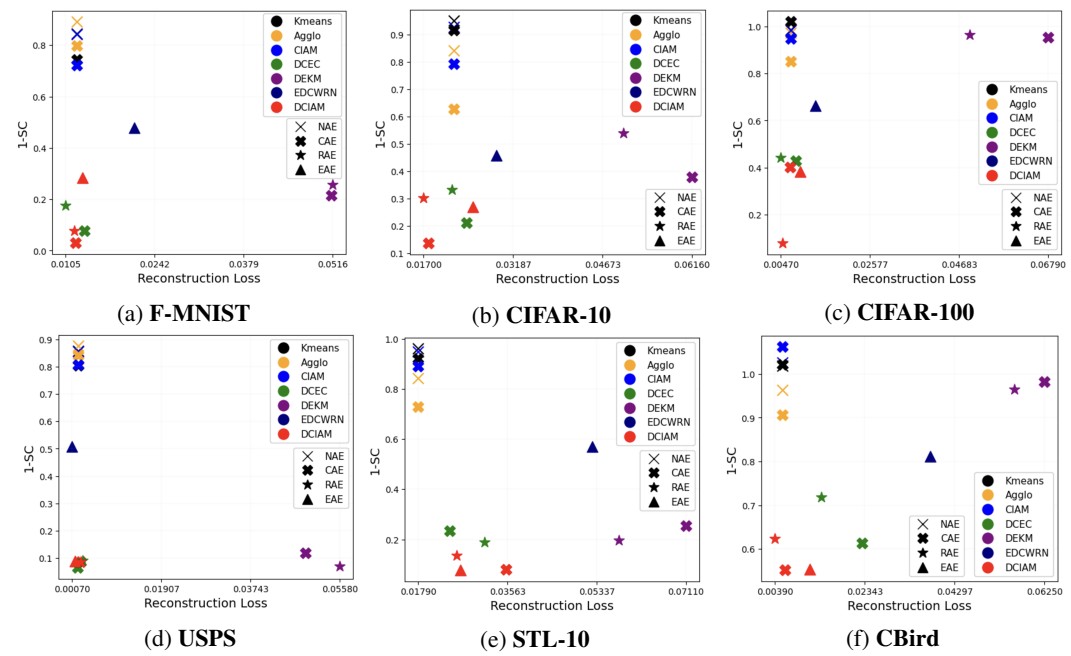

Figure 2: Reconstruction loss and clustering quality (1-SC) for all six image datasets. Different markers stand for various AE architectures, and different colors signify distinct methods. *Lower is better for both axes*, since we plot 1-SC on the $y$-axis.

displays the best NMI and its associated SC, RL, and cluster sizes. **DClAM** consistently outperforms traditional and deep clustering baselines in terms of all SC, RL and NMI metrics.

Table 4: SC for image datasets, comparing **DClAM** to baselines with different encoder/decoder architectures. Best for each dataset is in bold. See text for details. *Higher is better.*

| Dataset | Convolutional AE | | | ResNet AE | | | EAE | |
|---|---|---|---|---|---|---|---|---|
| | DCEC | DEKM | **DClAM** | DCEC | DEKM | **DClAM** | EDC | **DClAM** |
| FM | 0.923 | 0.785 | **0.970** | 0.824 | 0.742 | 0.922 | 0.521 | 0.715 |
| C-10 | 0.787 | 0.622 | **0.863** | 0.667 | 0.461 | 0.697 | 0.541 | 0.731 |
| C-100 | 0.572 | 0.047 | 0.598 | 0.557 | 0.036 | **0.921** | 0.337 | 0.636 |
| USPS | **0.935** | 0.882 | 0.914 | 0.909 | 0.843 | 0.914 | 0.491 | 0.911 |
| STL | 0.766 | 0.745 | 0.919 | 0.812 | 0.804 | 0.865 | 0.431 | **0.923** |
| CBird | 0.386 | 0.018 | **0.448** | 0.282 | 0.035 | 0.377 | 0.188 | 0.446 |

**Q2: Is DClAM's improvement agnostic to selected architecture?** Table 4 shows that the performance improvements achieved by **DClAM** is independent of the Autoencoder (AE) architecture choice. **DClAM** with *all three architectures* – CAE, EAE, and RAE – consistently outperforms their respective baselines, DCEC, DEKM and EDCWRN with similar architecture. That is, within each type of AE, **DClAM** has better results than DCEC and DEKM, or EDC. This not only underscores the superiority of the internal algorithm of **DClAM** over the corresponding baselines but also suggests the potential for further improvement with some more advanced AE architecture.

**Further results.** We qualitatively evaluate the clusters found by **DClAM** in Fig. 3 for Fashion MNIST (10 clusters) and Caltech Birds (10 out of 200 clusters), visualizing the learned memories (or cluster centers), and the corresponding closest and farthest cluster members (as measured in the latent space) from the centers. In most cases, the memories form an average image that match the closest images well. The farthest cluster members still appear similar to their memories in most cases, but do start changing significantly in some cases: (i) In the 2nd row (block 2) for FMNIST an image that looks like a pant is grouped with dresses though the overall image shape is still similar. (ii) In the 5th row

(block 1) for CBird, the memory and the closest image are very similar but the farthest image appears significantly different.

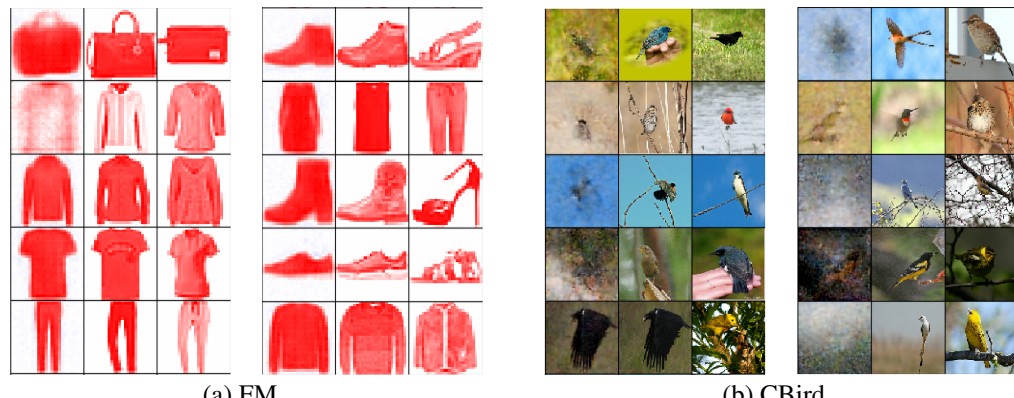

(a) FM                                        (b) CBird

Figure 3: Visualizing **DC1AM** clusters for (a) Fashion MNIST and (b) Caltech Birds, with the learned memories (left column in each block) and the corresponding closest (center column in block) and farthest (right column in block) images within their clusters.

In addition to the above, we discuss our thorough empirical evaluation in Appendix B, reporting various clustering metrics in Appendix B.1, and visualizing the evolution of the latent memories (cluster centers) in Appendix B.2.

## 6  LIMITATIONS AND FUTURE WORK

In this paper, we introduce a fresh integration of associative memories in a deep neural network module to create the innovative deep clustering algorithms **DC1AM** that leverages the AM attractor dynamics. Our findings demonstrate that **DC1AM** significantly surpasses standard prototype-based clustering and existing deep clustering methods. However, it is worth noting that **DC1AM** is still sensitive to hyperparameters and requires pretraining to avoid latent space collapse. Inspired by **DC1AM**'s outstanding performance, our future work aims to extend it to multimodal deep clustering. We plan to explore new energy functions and update dynamics to enhance spectral and semantic clustering. Having flexibility to add other encoder/decoder frameworks with **DC1AM**, we aim to explore transformer-based AE approaches in the future. Additionally, leveraging **DC1AM**'s flexibility, we intend to automate the estimation of the number of clusters directly from the data.

## REPRODUCIBILITY STATEMENT

We have included details of all the hyperparameter settings and other implementation details in Section 5 and in Appendix A. Our code will also be made available after the review period to facilitate reproducibility.

## ETHICS

This paper studies the core technical problem of unsupervised deep clustering. We do not anticipate any obvious issues of concern that need to be highlighted.

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

# A  EXPERIMENTAL DETAILS

## A.1  METRICS USED

To assess the performance of `DClAM`, we utilize the Silhouette Coefficient (SC) (Rousseeuw, 1987) as an unsupervised metric for measuring clustering quality. SC scores range from $-1$ to 1, where 1 indicates perfect clustering and $-1$ indicates completely incorrect labels. A score close to 0 suggests the existence of overlapping clusters. We also employ Normalized Mutual Information (NMI) (Vinh et al., 2009) to evaluate the alignment between the partition obtained by `DClAM` and the ground truth clustering labels. NMI scores range from 0 (completely incorrect) to 1 (perfect clustering). Additionally, we compute Reconstruction Loss (RL), representing the mean squared error between original and reconstructed points, where lower is better. Entropy (ETP) (Bein, 2006) and Cluster Size (CS) are computed to assess cluster balance. In clustering, higher entropy (the highest value is $\log_2(k)$ for each dataset, where $k$ is the number of true clusters) indicates more balanced clusters, while lower values suggest potential imbalance, possibly involving singleton or very small clusters. Entropy ($H(X)$) is calculated based on the distribution of data points across clusters:

$$H(X) = -\sum_{i=1}^{k} P(C_i) \ \log_2 P(C_i)$$

where, $P(C_i)$ is the proportion of data points in cluster $C_i$ relative to the total number of data points. Cluster Size (CS) indicates the largest and smallest clusters (in terms of the number of data points) identified in the dataset (more balanced clustering is better).

## A.2  PARAMETER SETTING

For Convolutional AE or CAE, for $k$-means, Agglomerative, `ClAM`, DCEC, DEKM, and `DClAM`, we adopt an architecture identical to DCEC. The encoder network structure follows $\text{conv}_{32}^5 \to \text{conv}_{64}^5 \to \text{conv}_{128}^3 \to \text{FC}_d$, where $\text{conv}_n^k$ represents a convolutional layer with $n$ filters and a kernel size of $k \times k$, and $\text{FC}_d$ denotes a fully connected layer of dimension $d$. Here, $d$ is the number of true clusters in the dataset, and serves as the latent dimension. The decoder mirrors the encoder.

The ResNet AE or RAE approach draws inspiration from the standard ResNet block described by Wickramasinghe et al. (2021). For DCEC, DEKM, and `DClAM` a streamlined configuration is employed using two filters with sizes 32 and 64. The size of the embedded representation is maintained at $d$, corresponding to the number of clusters in the dataset, as in the previous setup. In this experiment, the number of repeating layers in the ResNet block is set to 2. To enhance model performance, batch normalization and leakyReLU are incorporated. For a given number of repeats ($f$), the total number of hidden layers is calculated as 2 + ($f$ * number of filters), resulting in 6 layers in our case.

The EDCWRN AE or EAE, is that from Oskouei et al. (2023), so for both EDC and `DClAM`, we follow the proposed architecture, where the encoder network is configured as a fully connected multilayer perceptron (MLP) with dimensions $i$-500-500-2000-$d$ for all datasets, where $i$ represents the dimension of the input space (features), and $d$ is the number of clusters in the dataset. Similarly, the decoder network mirrors the encoder, constituting an MLP with dimensions $d$-2000-500-500-$i$. All internal layers, except for the input, output, and embedding layers, use the ReLU activation function.

All three architectures described above are pretrained end-to-end for 100 epochs using Adam (Kingma & Ba, 2014) with default parameters. The number of clusters $k$ is not a hyperparameter, but rather is taken as the true number of classes in each dataset. Also, as noted above, we set the latent dimensionality $d$ (or $m$) the same as the number of true classes $k$ in the dataset, i.e., $m = d = k$.

## A.3  IMPLEMENTATION DETAILS

We implement and evaluate `DClAM` using the Tensorflow (Abadi et al., 2016) library while employing `scikit-learn` (Pedregosa et al., 2011) for clustering baselines and quality metrics. We train our models on a single node with 1 NVIDIA RTX A6000 (48GB RAM) and a 16-core 2.4GHz Intel Xeon(R) Silver 4314 CPU. Hyperparameters are tuned individually for each dataset to maximize

the Silhouette Coefficient (Rousseeuw, 1987). Table 5 lists the hyperparameters, their roles, and respective values/ranges.

Table 5: Hyperparameters, their roles and range of values for `DClAM`.

| Hyperparameter | Used Values |
|---|---|
| Inverse temperature, $\beta$ | $[10^{-5}, ..., 5]$ |
| Number of layers, $T = \tau/dt$ | $[5, ..., 25]$ |
| Batch size | $[16, 32, 64, 128, 256]$ |
| Initial learning rate (AM), $\epsilon_{am}$ | $[10^{-4}, 10^{-3}, 10^{-2}, 10^{-1}]$ |
| Initial learning rate (AE), $\epsilon_{ae}$ | $[10^{-7}, 10^{-6}, 10^{-5}, 10^{-4}, 10^{-3}, 10^{-2}, 10^{-1}]$ |
| Reduce LR patience (epochs) | $[5, 10, 15]$ |

For baseline schemes like $k$-means and agglomerative, we use the `scikit-learn` library implementation, adjusting hyperparameters for optimal performance on each dataset. For DCEC (Guo et al., 2017b) and DEKM (Guo et al., 2021), we leverage their Tensorflow implementation[8] [9] and for EDCWRN (Oskouei et al., 2023), we utilze their Python implemtentation[10].

Table 6: Best hyperparameters for different datasets for `DClAM`. '-' denotes NA.

| Dataset | Inverse temperature, $\beta$ | | | Layers, $T$ | | | Batch size | | | Learning rate(AM) | | | Learning rate (e) | | | Learning rate (d) | | |
|---|---|---|---|---|---|---|---|---|---|---|---|---|---|---|---|---|---|---|
| | CAE | RAE | EAE | CAE | RAE | EAE | CAE | RAE | EAE | CAE | RAE | EAE | CAE | RAE | EAE | CAE | RAE | EAE |
| FM | 0.5 | 0.09 | 0.7 | 15 | 15 | 10 | 64 | 64 | 64 | 0.001 | 0.001 | 0.1 | 0.0000001 | 0.0000001 | 0.0000001 | 0.001 | 0.001 | 0.001 |
| C-10 | 2 | 0.02 | 0.5 | 15 | 15 | 12 | 64 | 64 | 64 | 0.001 | 0.001 | 0.01 | 0.0000001 | 0.0000001 | 0.0000001 | 0.001 | 0.001 | 0.001 |
| C-100 | 1 | 0.005 | 5 | 10 | 10 | 10 | 64 | 64 | 64 | 0.001 | 0.001 | 0.001 | 0.0000001 | 0.0000001 | 0.0000001 | 0.001 | 0.001 | 0.001 |
| USPS | 0.5 | 1 | 1 | 15 | 10 | 15 | 64 | 64 | 32 | 0.01 | 0.01 | 0.1 | 0.0000001 | 0.0000001 | 0.0000001 | 0.001 | 0.001 | 0.01 |
| STL | 0.5 | 0.003 | 0.1 | 15 | 10 | 12 | 64 | 64 | 128 | 0.001 | 0.01 | 0.1 | 0.0000001 | 0.0000001 | 0.0000001 | 0.001 | 0.001 | 0.0001 |
| CBird | 0.05 | 0.00015 | 0.005 | 15 | 10 | 15 | 64 | 64 | 64 | 0.01 | 0.001 | 0.1 | 0.0000001 | 0.0000001 | 0.0000001 | 0.001 | 0.001 | 0.001 |
| R-10K | - | - | 10 | - | - | 10 | - | - | 64 | - | - | 0.01 | - | - | 0.0000001 | - | - | 0.1 |
| 20-NG | - | - | 1.5 | - | - | 15 | - | - | 64 | - | - | 0.1 | - | - | 0.0000001 | - | - | 0.1 |

## A.4 HYPERPARAMETERS FOR `DClAM`

We extensively tune all hyperparameters (Table 5) for the optimal results in `DClAM`. We found that the inverse temperature $\beta$ serves as the most critical hyperparameter, which we explore in the range of $[10^{-5}, ..., 5]$ for tuning. We employ the Adam optimizer while keeping separate initial learning rates for the AM and AE networks. If the training loss does not improve for a certain number of epochs, we decrease the learning rate by a factor of 0.8 until it reaches the minimum threshold ($10^{-6}$). Each hyperparameter configuration is run mostly for 300 epochs (in certain cases longer training is needed for better results) with 5 restarts using different random seeds. Throughout each epoch, we track the training loss. The set of hyperparameters and the associated model yielding the lowest training loss are chosen during the inference step. The best hyperparameter values for various datasets for `DClAM` are detailed in Table 6.

## A.5 HYPERPARAMETERS FOR BASELINES

We compare `DClAM` with baseline clustering schemes $k$-means and agglomerative from `scikit-learn`, `ClAM`, DCEC, DEKM and EDCWRN. For $k$-means and agglomerative, we perform a comprehensive search for tuning different hyperparameters available in `scikit-learn` and pick the best results. For DCEC, DEKM and EDCWRN, we tuned all related hyperparams to obtain the best SC (for 10% RRL) and best RL (for 10% of best SC). For `ClAM` we precisely replicate the hyperparameter search criteria outlined in its respective paper, which closely aligns with our approach for `DClAM`, as detailed in Table 5. Table 7 provides a brief description of the hyperparameters and their roles in the baseline schemes.

---

[8]https://github.com/XifengGuo/DCEC

[9]https://github.com/spdj2271/DEKM/blob/main/DEKM.py

[10]https://github.com/Amin-Golzari-Oskouei/EDICWRN

Table 7: Hyperparameters (HPs), their roles and range of values for the baseline clustering schemes.

| Baseline | HP | Role | Used Values |
|---|---|---|---|
| $k$-means | init | Initialization method | ['k-means++', 'random'] |
| | n_init | Number of time the k-means algorithm will be run | 1000 |
| Agglomerative | affinity | Metric used to compute the linkage | ['euclidean', 'l1', 'l2', 'manhattan', 'cosine'] |
| | linkage | Linkage criterion to use | ['single', 'average', 'complete', 'ward'] |
| DCEC | batch_size | Size of each batch | [64, 128, 256] |
| | maxiter | Maximum number of iteration | [2e4, 3e4] |
| | alpha | Degree of freedom of student's t-distribution | 1 |
| | gamma | Coefficient of clustering loss | [0.01, 0.1, 1, 10] |
| | update_interval | Interval at which the predicted and target distributions are updated | [1, 2, 3, 4, 5, 10, 20, 30, 40, 50, 75, 100, 125, 140, 150, 200] |
| | tol | Tolerance rate | 0.001 |
| DEKM | batch_size | Size of each batch | [64, 128, 256] |
| | maxiter | Maximum number of iteration | 2e4 |
| | hidden_units | Number of latent dimension | Number of true cluster as per dataset |
| | update_interval | Interval at which the predicted and target distributions are updated | [10, 20, 30, 40, 50, 75, 100, 125, 140, 150, 200] |
| | tol | Tolerance rate | 0.000001 |
| EDCWRN | batch_size | Size of each batch | [64, 128, 256] |
| | maxiter_pretraining | Maximum number of iteration in pertaining | 500*batch_size |
| | maxiter_clustering | Maximum number of iteration in clustering | [8000, 16000, 24000] |
| | gamma | Coefficient of clustering loss | [0.01, 0.1, 1, 10] |
| | update_interval | Interval at which the predicted and target distributions are updated | [1, 2, 3, 4, 5, 10, 20, 30, 40, 50, 75, 100, 125, 140, 150, 200] |
| | tol | Tolerance rate | 0.0001 |
| **ClAM** | $\beta$ | Inverse temperature | $[10^{-5} - 5]$ |
| | $T = 1/\alpha = \tau/dt$ | Number of layers | [2-20] |
| | batch_size | Size of each batch | [8, 16, 32, 64, 128, 256] |
| | $\epsilon$ | Adam initial learning rate | $[10^{-4}, 10^{-3}, 10^{-2}, 10^{-1}]$ |
| | $\epsilon$_factor | Reduce LR by factor | 0.8 |
| | $\epsilon$_patience | Reduce LR patience (epochs) | 5 |
| | $\epsilon$_min | Minimum LR | $10^{-5}$ |
| | $\epsilon$_loss_threshold | Reduce LR loss threshold | $10^{-3}$ |
| | max_epochs | Maximum Number of epochs | 200 |
| | restart | Number of restart | 10 |
| | mask_prob | Mask probability | [0.1, 0.12, 0.15, 0.2, 0.25, 0.3] |
| | mask_val | Mask value | ['mean', 'min', 'max'] |

# B  ADDITIONAL EXPERIMENTAL RESULTS

## B.1  DETAILED RESULTS WITH VARIOUS CLUSTERING QUALITY METRICS

Table 8 provides a comprehensive overview of the metrics (SC, NMI, RL, ETP, and CS) for **DClAM**, and corresponding baselines, focusing on the best SC in each method across various AE architectures where RL is constrained to 10% of the pretrained AE loss. RL is not presented for $k$-means, Agglomerative and **ClAM** for the original space and for CAE as it remains consistent across the three methods after pre-taining. Similarly, Table 9 provides a similar overview of the metrics (SC, NMI, RL, ETP, and CS) for **DClAM**, and corresponding baselines, focusing on the best Relative RL (RRL) in each method across various AE architectures where SC is constrained to 10% of the best/peak SC of the method. Table 10 represents all corresponding metrics focusing on the best NMI in each method. These tables highlight that **DClAM** exhibits strong performance not only in terms of SC and RL, but also when compared to the ground truth labels via NMI. In fact, for NMI, **DClAM** has the best values in 5 out of the 8 datasets (DCEC has the best values on the other 3). Additionally, **DClAM** clusters maintain reasonable entropy (ETP) and cluster size (CS), ensuring a balanced clustering outcome.

For an understanding of the importance of ETP and CS in clustering, consider the case of Agglomerative clustering in the latent space (CAE) on the CIFAR-10 dataset (see Table 8). In this instance, almost all points (49991 out of 50000) belong to one cluster, while the other 9 clusters contain only one data point each, indicating very poor clustering. The low entropy (0.003) further highlights the deficiency of the clustering.

Table 8: **Metrics obtained by `DC1AM` and baselines corresponding to the best SC (RL within 10% of the pretrained AE loss).** The best performance for each dataset is in **boldface**. (note abbreviations DCEC→DC, EDCWRN→EDC, Entropy→ETP, Cluster-size→CS, No-AE→NAE, Conv-AE→CAE, EDCWRN-AE→EAE, Resnet-AE→RAE). '-' denotes NA. x▼ indicates negative RRL which means the RL of the method is x% less than the pretrained AE loss.

| Data | Met | Kmeans | | Agglo | | C1AM | | DC | | DEKM | | EDC | DC1AM | | |
|---|---|---|---|---|---|---|---|---|---|---|---|---|---|---|---|
| | | NAE | CAE | NAE | CAE | NAE | CAE | CAE | RAE | CAE | RAE | | CAE | EAE | RAE |
| FM | SC | 0.154 | 0.257 | 0.109 | 0.201 | 0.158 | 0.279 | 0.923 | 0.726 | 0.260 | 0.258 | 0.483 | **0.970** | 0.663 | 0.712 |
| | NMI | 0.511 | 0.643 | 0.534 | 0.624 | 0.521 | 0.622 | 0.558 | 0.624 | 0.551 | 0.609 | 0.495 | 0.472 | 0.511 | 0.488 |
| | RL | - | 0.0122 | - | 0.0122 | - | 0.0122 | 0.0134 | 0.0091 | 0.0105 | 0.0097 | 0.0096 | 0.0120 | 0.0096 | 0.0091 |
| | RRL | - | 0.0 | - | 0.0 | - | 0.0 | 9.8 | 9.6 | **13.9▼** | 18.1 | 10 | 1.6▼ | 10 | 9.6 |
| | ETP | 3.17 | 3.17 | 3.14 | 3.2 | 2.81 | 2.80 | 3.23 | 3.23 | 3.14 | 3.15 | 3.11 | 2.83 | 3.14 | 3.11 |
| | CS | 9617-2361 | 11145-2744 | 11830-1860 | 10298-2544 | 19032-1524 | 15679-2 | 10208-2733 | 8914-3338 | 12360-2310 | 10974-2724 | 12118-1478 | 20448-504 | 11734-2251 | 15906-2082 |
| C-10 | SC | 0.050 | 0.084 | 0.158 | 0.372 | 0.073 | 0.208 | 0.787 | 0.659 | 0.116 | 0.082 | 0.511 | **0.863** | 0.632 | 0.697 |
| | NMI | 0.078 | 0.122 | 0.0005 | 0.0004 | 0.073 | 0.015 | 0.074 | 0.094 | 0.123 | 0.129 | 0.112 | 0.075 | 0.061 | 0.079 |
| | RL | - | 0.0220 | - | 0.0220 | - | 0.0220 | 0.0241 | 0.0197 | 0.0239 | 0.0199 | 0.0184 | 0.0178 | 0.0184 | 0.0170 |
| | RRL | - | 0.0 | - | 0.0 | - | 0.0 | 9.6 | 8.9 | 8.6 | 11.1 | 10 | **19.5▼** | 10 | 5▼ |
| | ETP | 3.27 | 3.19 | 0.006 | 0.003 | 2.50 | 0.24 | 3.22 | 2.99 | 3.25 | 3.15 | 3.24 | 2.83 | 2.65 | 2.81 |
| | CS | 7105-2734 | 9779-2524 | 49979-1 | 49991-1 | 23544-582 | 48234-1 | 8511-2610 | 11229-1724 | 7905-3245 | 11731-2107 | 8198-2632 | 17521-425 | 13771-570 | 17121-569 |
| C-100 | SC | 0.015 | -0.020 | 0.028 | 0.149 | 0.018 | 0.053 | 0.314 | 0.470 | -0.007 | -0.016 | 0.311 | **0.598** | 0.536 | 0.482 |
| | NMI | 0.161 | 0.183 | 0.036 | 0.004 | 0.153 | 0.156 | 0.104 | 0.119 | 0.238 | 0.184 | 0.181 | 0.110 | 0.202 | 0.125 |
| | RL | - | 0.0070 | - | 0.0070 | - | 0.0070 | 0.0059 | 0.0043 | 0.0046 | 0.0041 | 0.0106 | 0.0069 | 0.0099 | 0.0044 |
| | RRL | - | 0.0 | - | 0.0 | - | 0.0 | **15.7▼** | 7.5 | **34.3▼** | 2.5 | 4.3 | 1.4▼ | 3.1 | 10 |
| | ETP | 6.53 | 6.48 | 0.940 | 0.052 | 6.51 | 4.38 | 5.54 | 4.52 | 6.25 | 6.46 | 6.49 | 4.16 | 5.85 | 4.03 |
| | CS | 1160-129 | 1395-23 | 38814-1 | 49834-1 | 1317-177 | 13950-11 | 2255-325 | 12721-122 | 1715-5 | 1322-87 | 999-216 | 11085-112 | 4116-32 | 8245-112 |
| USPS | SC | 0.143 | 0.195 | 0.124 | 0.158 | 0.144 | 0.194 | **0.935** | 0.758 | 0.195 | 0.217 | 0.461 | 0.820 | 0.872 | 0.891 |
| | NMI | 0.573 | 0.628 | 0.627 | 0.680 | 0.475 | 0.619 | 0.732 | 0.761 | 0.631 | 0.665 | 0.467 | 0.444 | 0.347 | 0.428 |
| | RL | - | 0.0019 | - | 0.0019 | - | 0.0019 | 0.0018 | 0.0019 | 0.0020 | 0.0024 | 0.0005 | 0.0021 | 0.0006 | 0.0025 |
| | RRL | - | 0.0 | - | 0.0 | - | 0.0 | 5.3▼ | 17.4▼ | 5.3 | 4.3 | 0.0 | 10 | 10 | 8.7 |
| | ETP | 3.27 | 3.23 | 3.26 | 3.27 | 3.10 | 3.16 | 3.26 | 3.29 | 3.23 | 3.25 | 3.29 | 3.12 | 2.78 | 2.99 |
| | CS | 284-121 | 359-89 | 333-121 | 328-104 | 420-53 | 375-64 | 287-106 | 281-138 | 379-90 | 319-99 | 295-134 | 442-71 | 841-76 | 524-49 |
| STL | SC | 0.039 | 0.079 | 0.158 | 0.270 | 0.051 | 0.108 | 0.132 | 0.259 | 0.082 | 0.081 | 0.411 | 0.475 | **0.891** | 0.615 |
| | NMI | 0.127 | 0.152 | 0.007 | 0.004 | 0.106 | 0.139 | 0.180 | 0.162 | 0.167 | 0.167 | 0.066 | 0.077 | 0.073 | 0.119 |
| | RL | - | 0.0179 | - | 0.0179 | - | 0.0179 | 0.0204 | 0.0189 | 0.0180 | 0.0174 | 0.0196 | 0.0187 | 0.0227 | 0.0190 |
| | RRL | - | 0.0 | - | 0.0 | - | 0.0 | 13.9 | 9.2 | 0.6 | 0.6 | 4.9▼ | 4.5 | 10 | 9.8 |
| | ETP | 3.26 | 3.25 | 0.069 | 0.025 | 2.43 | 1.4 | 3.19 | 3.17 | 3.21 | 3.19 | 2.92 | 2.48 | 2.99 | 2.87 |
| | CS | 764-312 | 830-287 | 4969-1 | 4991-1 | 2586-82 | 3888-38 | 931-239 | 1003-263 | 844-191 | 804-258 | 2611-33 | 2076-21 | 912-45 | 1219-113 |
| CBird | SC | -0.019 | -0.021 | 0.037 | 0.094 | -0.026 | -0.062 | 0.311 | 0.251 | -0.032 | -0.037 | 0.171 | **0.448** | 0.446 | 0.312 |
| | NMI | 0.412 | 0.353 | 0.206 | 0.132 | 0.423 | 0.485 | 0.347 | 0.299 | 0.372 | 0.370 | 0.471 | 0.221 | 0.467 | 0.211 |
| | RL | - | 0.0055 | - | 0.0055 | - | 0.0055 | 0.0061 | 0.0040 | 0.0055 | 0.0036 | 0.0206 | 0.0060 | 0.0115 | 0.0039 |
| | RRL | - | 0.0 | - | 0.0 | - | 0.0 | 10 | 10 | 0.0 | 0.0 | 10 | 9.1 | **39▼** | 8.3 |
| | ETP | 6.34 | 5.59 | 2.71 | 0.958 | 6.56 | 7.21 | 5.41 | 5.04 | 5.81 | 5.80 | 7.41 | 5.68 | 7.02 | 5.07 |
| | CS | 131-1 | 245-1 | 1722-1 | 2773-1 | 101-2 | 99-2 | 241-1 | 291-1 | 168-1 | 197-1 | 37-2 | 213-1 | 99-1 | 676-1 |
| R-10k | SC | -0.010 | | 0.114 | | -0.002 | | - | - | - | - | 0.023 | - | **0.564** | - |
| | NMI | 0.398 | | 0.012 | | 0.383 | | - | - | - | - | 0.152 | - | 0.367 | - |
| | RL | - | | - | | - | | - | - | - | - | 0.0011 | - | 0.0011 | - |
| | RRL | - | | - | | - | | - | - | - | - | 10 | - | 10 | - |
| | ETP | 5.13 | | 0.072 | | 5.10 | | - | - | - | - | 5.51 | - | 4.77 | - |
| | CS | 916-20 | | 11172-1 | | 885-18 | | - | - | - | - | 721-51 | - | 1046-1 | - |
| 20NG | SC | -0.021 | | 0.114 | | -0.008 | | - | - | - | - | 0.101 | - | **0.197** | - |
| | NMI | 0.155 | | 0.003 | | 0.166 | | - | - | - | - | 0.019 | - | 0.181 | - |
| | RL | - | | - | | - | | - | - | - | - | 0.0009 | - | 0.0009 | - |
| | RRL | - | | - | | - | | - | - | - | - | 10 | - | 10 | - |
| | ETP | 4.03 | | 0.022 | | 3.86 | | - | - | - | - | 4.32 | - | 4.21 | - |
| | CS | 2217-107 | | 18818-1 | | 3428-26 | | - | - | - | - | 1131-599 | - | 1812-199 | - |

In certain situations, when comparing two clustering methods, it can happen that a method performs better in terms of SC and RL but still exhibits a lower NMI compared to another method (see Table 8 for USPS where `DC1AM` outperforms DCEC in both CAE and RAE architecture in both SC and RL, however, the NMI is worse than DCEC in both cases). This indicates that the alignment of semantic class (ground truth or true underlying structure) with the geometric characteristics of the data might not be consistent or straightforward.

## B.2    HOW INTERPRETABLE ARE THE MEMORIES OF `DC1AM`?

We explore the prototype-based representation of the learned memories in latent space for `DC1AM` for Fashion-MNIST and USPS in Fig. 4. For Fashion-MNIST, the 60k images are partitioned into 10 clusters, and the evolution of memories is visualized in Fig. 4a during the training process outlined in Algorithm 1 for `DC1AM`. In each sub-figure we observe the evolution over epochs. At epoch 0, there are no distinct memories for clustering; instead, there are pairs of pullover (rows 3 and 5), shirts (rows 7 and 8), and t-shirts/tops (rows 6 and 9). However, discernible patterns emerge at epoch 10, refining further by epoch 20. By epoch 100, all ten memories represent distinct shapes, representing different cluster centroids.

Table 9: **Metrics obtained by `DClAM` and baselines corresponding to the best RL (SC within 10% of the best SC of the method).** The best performance for each dataset is in **boldface**. (note abbreviations DCEC→DC, EDCWRN→EDC, Entropy→ETP, Cluster-size→CS, No-AE→NAE, Conv-AE→CAE, EDCWRN-AE→EAE, Resnet-AE→RAE). '-' denotes NA. x▼ indicates negative RRL which means the RL of the method is x% less than the pretrained AE loss.

| Data | Met | Kmeans | | Agglo | | ClAM | | DC | | DEKM | | EDC | DClAM | | |
|---|---|---|---|---|---|---|---|---|---|---|---|---|---|---|---|
| | | NAE | CAE | NAE | CAE | NAE | CAE | CAE | RAE | CAE | RAE | | CAE | EAE | RAE |
| FM | SC | 0.154 | 0.257 | 0.109 | 0.201 | 0.158 | 0.279 | 0.898 | 0.824 | 0.785 | 0.742 | 0.521 | 0.891 | 0.715 | **0.922** |
| | NMI | 0.511 | 0.643 | 0.534 | 0.624 | 0.521 | 0.622 | 0.561 | 0.623 | 0.571 | 0.633 | 0.493 | 0.472 | 0.522 | 0.401 |
| | RL | - | 0.0122 | - | 0.0122 | - | 0.0122 | 0.0109 | 0.0105 | 0.0514 | 0.0516 | 0.0211 | 0.0102 | 0.0131 | 0.0118 |
| | RRL | - | 0.0 | - | 0.0 | - | 0.0 | 9.8▼ | 26.5 | 321 | 522 | 143 | **16.4▼** | 54.0 | 42.2 |
| | ETP | 3.17 | 3.17 | 3.14 | 3.2 | 2.81 | 2.80 | 3.21 | 3.6 | 3.15 | 3.16 | 3.09 | 2.83 | 3.16 | 2.98 |
| | CS | 9617-2361 | 11145-2744 | 11830-1860 | 10298-2544 | 19032-1524 | 15679-2 | 11307-2766 | 9450-3132 | 12720-2478 | 11178-2658 | 13199-1391 | 17040-504 | 11886-2148 | 11830-1290 |
| C-10 | SC | 0.050 | 0.084 | 0.158 | 0.372 | 0.073 | 0.208 | 0.786 | 0.667 | 0.622 | 0.461 | 0.541 | **0.809** | 0.731 | 0.697 |
| | NMI | 0.078 | 0.122 | 0.0005 | 0.0004 | 0.073 | 0.015 | 0.099 | 0.094 | 0.092 | 0.119 | 0.111 | 0.079 | 0.060 | 0.082 |
| | RL | - | 0.0220 | - | 0.0220 | - | 0.0220 | 0.0217 | 0.0217 | 0.0616 | 0.0502 | 0.0291 | 0.0175 | 0.0252 | 0.0171 |
| | RRL | - | 0.0 | - | 0.0 | - | 0.0 | 0.9▼ | 20 | 180 | 179 | 74.3 | **20.4▼** | 50.9 | 5▼ |
| | ETP | 3.27 | 3.19 | 0.006 | 0.003 | 2.50 | 0.24 | 3.15 | 2.99 | 2.01 | 3.07 | 3.25 | 2.83 | 2.64 | 2.50 |
| | CS | 7105-2734 | 9779-2524 | 49979-1 | 49991-1 | 23544-582 | 48234-1 | 7145-4055 | 11025-1542 | 23420-26 | 14530-2505 | 8172-2562 | 17520-390 | 14890-120 | 17121-455 |
| C-100 | SC | 0.015 | -0.020 | 0.028 | 0.149 | 0.018 | 0.053 | 0.572 | 0.557 | 0.047 | 0.036 | 0.337 | 0.540 | 0.617 | **0.921** |
| | NMI | 0.161 | 0.183 | 0.036 | 0.004 | 0.153 | 0.156 | 0.158 | 0.119 | 0.162 | 0.221 | 0.186 | 0.112 | 0.201 | 0.094 |
| | RL | - | 0.0070 | - | 0.0070 | - | 0.0070 | 0.0083 | 0.0047 | 0.0679 | 0.0494 | 0.0128 | 0.0061 | 0.0092 | 0.0051 |
| | RRL | - | 0.0 | - | 0.0 | - | 0.0 | 18.6 | 17.5 | 870 | 1135 | 33.3 | **12.9▼** | 4.2▼ | 27.5 |
| | ETP | 6.53 | 6.48 | 0.940 | 0.052 | 6.51 | 4.38 | 5.8 | 4.06 | 6.18 | 6.11 | 4.02 | 4.02 | 5.83 | 3.22 |
| | CS | 1160-129 | 1395-23 | 38814-1 | 49834-1 | 1317-177 | 13950-11 | 2540-115 | 13736-12 | 1950-10 | 1980-10 | 996-156 | 11010-25 | 4350-10 | 22480-1 |
| USPS | SC | 0.143 | 0.195 | 0.124 | 0.158 | 0.144 | 0.194 | **0.929** | 0.909 | 0.882 | 0.843 | 0.491 | 0.914 | 0.911 | 0.914 |
| | NMI | 0.573 | 0.628 | 0.627 | 0.680 | 0.475 | 0.619 | 0.717 | 0.736 | 0.691 | 0.684 | 0.451 | 0.477 | 0.339 | 0.437 |
| | RL | - | 0.0019 | - | 0.0019 | - | 0.0019 | 0.0014 | 0.0029 | 0.0487 | 0.0558 | 0.0007 | 0.0025 | 0.0013 | 0.0025 |
| | RRL | - | 0.0 | - | 0.0 | - | 0.0 | 26.3▼ | 26.1 | 2463 | 2326 | 40 | 31.6 | 160 | 8.7 |
| | ETP | 3.27 | 3.23 | 3.26 | 3.27 | 3.10 | 3.16 | 3.27 | 3.27 | 3.24 | 3.25 | 3.29 | 3.11 | 2.55 | 2.99 |
| | CS | 284-121 | 359-89 | 333-121 | 328-104 | 420-53 | 375-64 | 283-106 | 283-127 | 334-87 | 312-97 | 294-156 | 463-35 | 947-27 | 513-49 |
| STL | SC | 0.039 | 0.079 | 0.158 | 0.270 | 0.051 | 0.108 | 0.766 | 0.812 | 0.745 | 0.804 | 0.431 | 0.919 | **0.923** | 0.865 |
| | NMI | 0.127 | 0.152 | 0.007 | 0.004 | 0.106 | 0.139 | 0.181 | 0.170 | 0.149 | 0.152 | 0.065 | 0.144 | 0.072 | 0.107 |
| | RL | - | 0.0179 | - | 0.0179 | - | 0.0179 | 0.0242 | 0.0310 | 0.0711 | 0.0578 | 0.0525 | 0.0354 | 0.0263 | 0.0255 |
| | RRL | - | 0.0 | - | 0.0 | - | 0.0 | 35.8 | 79.2 | 297 | 234 | 155 | 97.8 | 27.7 | 47.4 |
| | ETP | 3.26 | 3.25 | 0.069 | 0.025 | 2.43 | 1.4 | 3.23 | 3.26 | 3.24 | 3.22 | 2.90 | 2.48 | 2.98 | 2.86 |
| | CS | 764-312 | 830-287 | 4969-1 | 4991-1 | 2586-82 | 3888-38 | 725-229 | 741-299 | 4064-16 | 821-261 | 2641-23 | 2280-27 | 929-34 | 1466-69 |
| CBird | SC | -0.019 | -0.021 | 0.037 | 0.094 | -0.026 | -0.062 | 0.386 | 0.282 | 0.018 | 0.035 | 0.188 | 0.413 | **0.441** | 0.377 |
| | NMI | 0.412 | 0.353 | 0.206 | 0.132 | 0.423 | 0.485 | 0.333 | 0.297 | 0.316 | 0.273 | 0.484 | 0.222 | 0.466 | 0.209 |
| | RL | - | 0.0055 | - | 0.0055 | - | 0.0055 | 0.0229 | 0.0139 | 0.0625 | 0.0560 | 0.0377 | 0.0056 | 0.0104 | 0.0039 |
| | RRL | - | 0.0 | - | 0.0 | - | 0.0 | 316 | 286 | 1036 | 1455 | 102 | 1.8 | 44.4▼ | 8.3 |
| | ETP | 6.34 | 5.68 | 2.71 | 0.958 | 6.56 | 7.21 | 5.51 | 5.03 | 5.16 | 4.47 | 7.43 | 5.68 | 7.01 | 5.06 |
| | CS | 131-1 | 245-1 | 1722-1 | 2773-1 | 101-2 | 99-2 | 248-1 | 297-1 | 312-1 | 519-1 | 35-2 | 211-1 | 100-1 | 701-1 |
| R-10k | SC | -0.010 | - | 0.114 | - | -0.002 | - | - | - | - | - | 0.035 | - | **0.673** | - |
| | NMI | 0.398 | - | 0.012 | - | 0.383 | - | - | - | - | - | 0.147 | - | 0.378 | - |
| | RL | - | - | - | - | - | - | - | - | - | - | 0.0016 | - | 0.0022 | - |
| | RRL | - | - | - | - | - | - | - | - | - | - | **60** | - | 120 | - |
| | ETP | 5.13 | - | 0.072 | - | 5.10 | - | - | - | - | - | 5.55 | - | 4.79 | - |
| | CS | 916-20 | - | 11172-1 | - | 885-18 | - | - | - | - | - | 727-56 | - | 1026-1 | - |
| 20NG | SC | -0.021 | - | 0.114 | - | -0.008 | - | - | - | - | - | 0.099 | - | **0.287** | - |
| | NMI | 0.155 | - | 0.003 | - | 0.166 | - | - | - | - | - | 0.018 | - | 0.180 | - |
| | RL | - | - | - | - | - | - | - | - | - | - | 0.0006 | - | 0.0012 | - |
| | RRL | - | - | - | - | - | - | - | - | - | - | **25▼** | - | 50 | - |
| | ETP | 4.03 | - | 0.022 | - | 3.86 | - | - | - | - | - | 4.31 | - | 4.19 | - |
| | CS | 2217-107 | - | 18818-1 | - | 3428-26 | - | - | - | - | - | 1142-582 | - | 1809-197 | - |

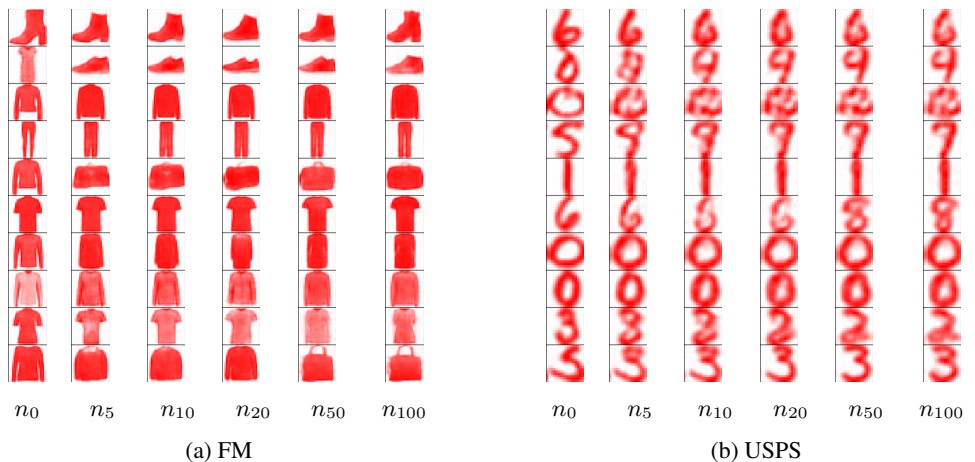

|   |   |   |   |   |   |
|---|---|---|---|---|---|
| $n_0$ | $n_5$ | $n_{10}$ | $n_{20}$ | $n_{50}$ | $n_{100}$ |

(a) FM    (b) USPS

Figure 4: **Evolution of prototypes for Fashion-MNIST and USPS for `DClAM`.** We visualize the prototypes at the $n^{\text{th}}$ training epoch for $n = 0, 5, 10, 20, 50, 100$ (with $T = 10$).

Table 10: **Metrics obtained by `DC1AM` and baselines corresponding to the best NMI.** The best performance for each dataset is in **boldface**. (note abbreviations DCEC→DC, EDCWRN→EDC, Entropy→ETP, Cluster-size→CS, No-AE→NAE, Conv-AE→CAE, EDCWRN-AE→EAE, Resnet-AE→RAE). '-' denotes NA. x▼ indicates negative RRL which means the RL of the method is x% less than the pretrained AE loss.

| Data | Met | Kmeans NAE | Kmeans CAE | Agglo NAE | Agglo CAE | C1AM NAE | C1AM CAE | DC CAE | DC RAE | DEKM CAE | DEKM RAE | EDC | DC1AM CAE | DC1AM EAE | DC1AM RAE |
|---|---|---|---|---|---|---|---|---|---|---|---|---|---|---|---|
| FM | SC | 0.154 | 0.251 | 0.109 | 0.201 | 0.140 | 0.262 | 0.861 | 0.716 | 0.819 | 0.784 | 0.430 | 0.817 | 0.619 | 0.825 |
| | NMI | 0.511 | 0.643 | 0.534 | 0.625 | 0.525 | 0.631 | 0.629 | **0.668** | 0.586 | 0.639 | 0.457 | 0.610 | 0.534 | 0.597 |
| | RL | - | 0.0122 | - | 0.0122 | - | 0.0122 | 0.0138 | 0.0139 | 0.0574 | 0.0596 | 0.0263 | 0.0406 | 0.0327 | 0.0387 |
| | RRL | - | 0.0 | - | 0.0 | - | 0.0 | 13.1 | 67.5 | 370 | 618 | 202 | 233 | 276 | 366 |
| | ETP | 3.17 | 3.17 | 3.14 | 3.20 | 3.13 | 2.98 | 3.22 | 3.20 | 3.07 | 3.16 | 3.00 | 3.16 | 3.22 | 3.18 |
| | CS | 9617-2361 | 11145-2744 | 11830-1860 | 10298-2544 | 14068-2435 | 15262-2100 | 10886-3030 | 9734-2847 | 12974-1191 | 11023-2652 | 17140-1578 | 11028-2658 | 10332-3054 | 10404-2610 |
| C-10 | SC | 0.050 | 0.072 | 0.014 | 0.020 | 0.064 | 0.101 | 0.118 | 0.653 | 0.276 | 0.262 | 0.541 | 0.713 | 0.632 | 0.420 |
| | NMI | 0.078 | 0.122 | 0.071 | 0.101 | 0.086 | 0.105 | 0.121 | 0.120 | 0.116 | 0.122 | 0.111 | **0.123** | 0.114 | 0.119 |
| | RL | - | 0.0220 | - | 0.0220 | - | 0.0220 | 0.0221 | 0.0245 | 0.0426 | 0.0362 | 0.0291 | 0.0403 | 0.0379 | 0.0326 |
| | RRL | - | 0.0 | - | 0.0 | - | 0.0 | 0.5 | 36.1 | 93.6 | 101 | 74.3 | 83.2 | 127 | 81.1 |
| | ETP | 3.27 | 3.19 | 3.17 | 3.02 | 3.23 | 2.21 | 3.07 | 3.21 | 3.19 | 3.11 | 3.25 | 3.18 | 2.98 | 3.28 |
| | CS | 7105-2734 | 9779-2524 | 10505-1650 | 11278-1764 | 9587-2925 | 26395-361 | 11022-3374 | 10235-1968 | 13746-2168 | | 8172-2562 | 8595-2365 | 10721-289 | 6843-3144 |
| C-100 | SC | 0.015 | -0.014 | -0.018 | -0.043 | 0.018 | 0.001 | 0.048 | 0.002 | -0.011 | -0.028 | 0.308 | 0.354 | 0.200 | 0.130 |
| | NMI | 0.161 | 0.183 | 0.150 | 0.167 | 0.153 | 0.170 | 0.162 | 0.179 | 0.186 | 0.189 | 0.186 | 0.219 | 0.225 | **0.239** |
| | RL | - | 0.0070 | - | 0.0070 | - | 0.0070 | 0.0072 | 0.0049 | 0.0112 | 0.0074 | 0.0398 | 0.0257 | 0.0250 | 0.0226 |
| | RRL | - | 0.0 | - | 0.0 | - | 0.0 | 2.9 | 22.5 | 60 | 85 | 315 | 267 | 160 | 465 |
| | ETP | 6.53 | 6.48 | 6.45 | 6.30 | 6.51 | 6.27 | 6.41 | 6.41 | 5.23 | 6.45 | 5.51 | 6.33 | 6.33 | 6.36 |
| | CS | 1160-129 | 1395-23 | 1299-77 | 2308-17 | 1317-177 | 2535-39 | 1623-14 | 1380-21 | 2213-32 | 1440-60 | 996-156 | 1210-5 | 2105-15 | 1315-5 |
| USPS | SC | 0.143 | 0.195 | 0.124 | 0.159 | 0.142 | 0.180 | 0.920 | 0.896 | 0.946 | 0.465 | 0.43 | 0.865 | 0.660 | 0.857 |
| | NMI | 0.573 | 0.628 | 0.627 | 0.680 | 0.564 | 0.640 | **0.737** | 0.736 | 0.728 | 0.701 | 0.689 | 0.689 | 0.583 | 0.660 |
| | RL | - | 0.0019 | - | 0.0019 | - | 0.0019 | 0.0074 | 0.0039 | 0.0748 | 0.0374 | 0.0006 | 0.0451 | 0.0322 | 0.0409 |
| | RRL | - | 0.0 | - | 0.0 | - | 0.0 | 289 | 69.6 | 3836 | 1526 | 20 | 2274 | 6340 | 1678 |
| | ETP | 3.27 | 3.23 | 3.26 | 3.27 | 3.27 | 3.21 | 3.27 | 3.27 | 3.24 | 3.24 | 3.29 | 3.11 | 3.24 | 3.23 |
| | CS | 284-121 | 359-89 | 333-121 | 328-100 | 290-132 | 343-73 | 289-80 | 282-107 | 298-80 | 318-91 | 294-156 | 396-35 | 385-107 | 308-72 |
| STL | SC | 0.039 | 0.074 | 0.024 | 0.021 | 0.042 | 0.069 | 0.822 | 0.837 | 0.109 | 0.079 | 0.332 | 0.388 | 0.597 | 0.280 |
| | NMI | 0.127 | 0.152 | 0.121 | 0.138 | 0.130 | 0.169 | **0.188** | 0.165 | 0.170 | 0.166 | 0.103 | 0.149 | 0.151 | 0.159 |
| | RL | - | 0.0179 | - | 0.0179 | - | 0.0179 | 0.0328 | 0.0362 | 0.0315 | 0.0174 | 0.0433 | 0.0409 | 0.0454 | 0.0364 |
| | RRL | - | 0.0 | - | 0.0 | - | 0.0 | 83.2 | 109 | 76.0 | 0.6 | 110 | 128 | 120 | 110 |
| | ETP | 3.26 | 3.25 | 3.02 | 3.02 | 3.24 | 2.82 | 3.24 | 3.28 | 3.21 | 3.20 | 2.62 | 3.13 | 3.18 | 3.15 |
| | CS | 764-312 | 830-287 | 1379-205 | 1373-130 | 945-317 | 1212-2 | 849-232 | 671-326 | 807-250 | 876-264 | 2173-121 | 982-46 | 929-232 | 938-181 |
| CBird | SC | -0.019 | -0.021 | -0.018 | -0.064 | -0.026 | -0.062 | 0.248 | 0.152 | -0.041 | -0.038 | 0.188 | 0.135 | 0.068 | 0.167 |
| | NMI | 0.412 | 0.353 | 0.469 | 0.439 | 0.423 | 0.485 | 0.356 | 0.320 | 0.364 | 0.370 | 0.484 | 0.421 | **0.493** | 0.385 |
| | RL | - | 0.0055 | - | 0.0055 | - | 0.0055 | 0.0229 | 0.0152 | 0.0066 | 0.0036 | 0.0377 | 0.0255 | 0.0237 | 0.0249 |
| | RRL | - | 0.0 | - | 0.0 | - | 0.0 | 316 | 322 | | | 102 | | | |
| | ETP | 6.34 | | 6.97 | | 6.56 | | 5.84 | 5.12 | 5.71 | 5.80 | 7.43 | 6.48 | 7.39 | 6.05 |
| | CS | 131-1 | 245-1 | 93-1 | 232-1 | 101-2 | 99-2 | 167-1 | 570-1 | 177-1 | 197-1 | 35-2 | 143-1 | 58-2 | 180-1 |
| R-10k | SC | -0.010 | - | -0.012 | - | -0.007 | - | - | - | - | - | 0.013 | - | 0.647 | - |
| | NMI | 0.398 | - | 0.404 | - | 0.394 | - | - | - | - | - | 0.169 | - | **0.414** | - |
| | RL | - | - | - | - | - | - | - | - | - | - | 0.0014 | - | 0.0020 | - |
| | RRL | - | - | - | - | - | - | - | - | - | - | 40 | - | 100 | - |
| | ETP | 5.13 | - | 5.15 | - | 5.22 | - | - | - | - | - | 5.47 | - | 5.2 | - |
| | CS | 916-20 | - | 845-18 | - | 650-41 | - | - | - | - | - | 478-76 | - | 540-1 | - |
| 20NG | SC | -0.021 | - | -0.186 | - | -0.103 | - | - | - | - | - | 0.066 | - | 0.199 | - |
| | NMI | 0.155 | - | 0.167 | - | 0.176 | - | - | - | - | - | 0.018 | - | **0.229** | - |
| | RL | - | - | - | - | - | - | - | - | - | - | 0.0006 | - | 0.0012 | - |
| | RRL | - | - | - | - | - | - | - | - | - | - | 25▼ | - | 50 | - |
| | ETP | 4.03 | - | 3.64 | - | 3.77 | - | - | - | - | - | 4.31 | - | 3.87 | - |
| | CS | 2217-107 | - | 4024-52 | - | 4227-103 | - | - | - | - | - | 1142-582 | - | 3203-105 | - |

## C  Additional Experiments and Results

### C.1  Pretrained losses for all architecture and all datasets

Table 11 records the pretrained reconstruction losses (RL) for all architectures and all datasets. These are the base RL values $RL_p$ used when computing RRL in Eq. (11).

Table 11: Per-dataset, per-architecture pretrained loss. Note abbreviations Conv-AE→CAE, Resnet-AE→RAE, and EDCWRN-AE→EAE. Further, '-' denotes NA.

| Dataset | Architecture | | |
|---|---|---|---|
| | CAE | RAE | EAE |
| FM | 0.0122 | 0.0083 | 0.0087 |
| C-10 | 0.0220 | 0.0180 | 0.0167 |
| C-100 | 0.0070 | 0.0040 | 0.0096 |
| USPS | 0.0019 | 0.0023 | 0.0005 |
| STL | 0.0179 | 0.0173 | 0.0206 |
| CBird | 0.0055 | 0.0036 | 0.0187 |
| R-10K | - | - | 0.0010 |
| 20NG | - | - | 0.0008 |

### C.2  Additional details on hyperparameter selection

In Figs. 5 to 9, we plot the reconstruction loss (RL) and the silhouette coefficient (SC) for each hyperparameter configuration considered for **DC1AM** and the baselines DCEC and DEKM for the different vision datasets (reported in Tables 2, 3, 4, 8, and 9). We also highlight the *Pareto front* for each of the dataset/method pairs, and the dotted vertical and horizontal lines denote the RL and (1-SC) values corresponding to the 10% margin from the best RL and (1-SC). Furthermore, the red and cyan highlighted points show the best hyperparameter configuration corresponding to the metric reported in Table 8 and 9. These results clearly highlight how we thoroughly optimize the hyperparameters, and how we select the final Pareto optimal performance values from the Pareto front to be consistent and fair across all methods. Some of the results in Table 8 and 9 have been updated based on our extended exploration of the Pareto front of hyperparameters based on reviewer comments. The overall trend and performance are in accord with our main claim, namely that **DC1AM** offers the best clustering performance in terms of SC, as well as having low reconstruction loss. It also performs very well on the supervised NMI metric. In fact, for NMI, **DC1AM** has the best value in 5 out of the 8 datasets (see Table 10).

### C.3  Effect of latent dimensionality

In Figure 10, we demonstrate the effect of varying the dimensionality $m$ of the latent space for USPS dataset, which has 10 classes. We can see that $m$=10 has a good trade-off between RL and SC compared to other values which strengthens our motivation to set the latent dimension as equal to the number of clusters.

### C.4  Ablating the number of AM layers $T$

In Figure 11, we ablate the effect of $T$, the number of AM layers. We see that $T$ controls the trade-off between RL and SC, with high values of $T$ leading to high SC. However, we see there are moderate values of $T$ where are able to obtain high values of SC while maintaining low RL (after sufficient number of epochs of training), and varying the value of $T$ allows us to explore the Pareto front of RL and SC.

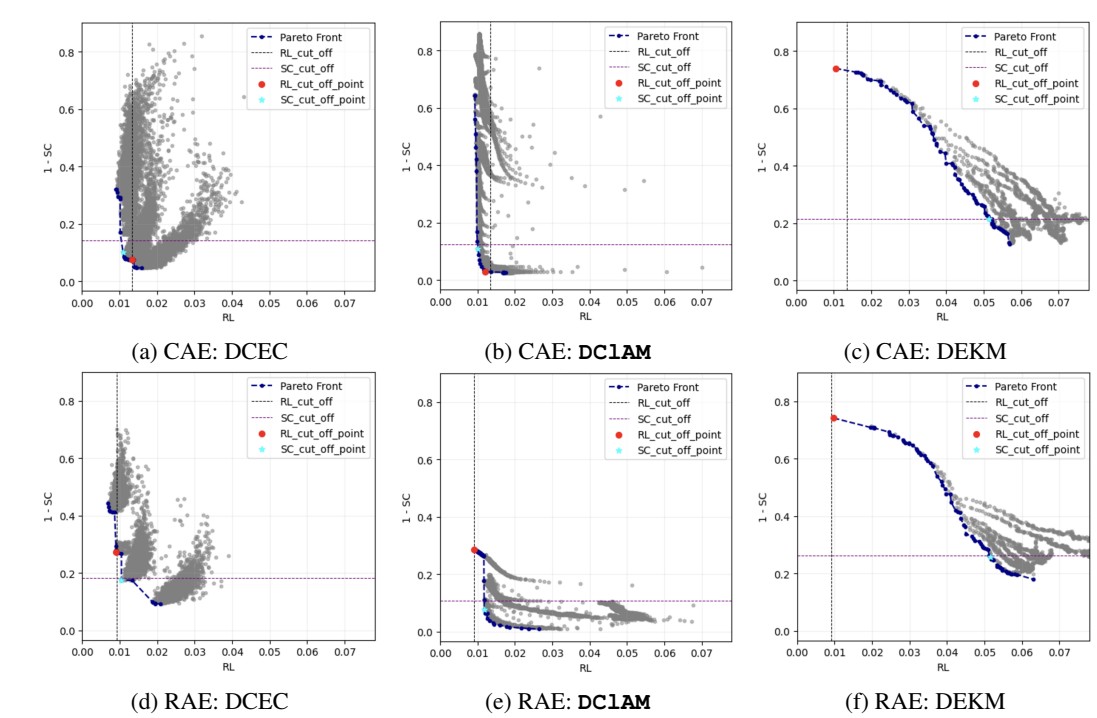

(a) CAE: DCEC  (b) CAE: **DC1AM**  (c) CAE: DEKM

(d) RAE: DCEC  (e) RAE: **DC1AM**  (f) RAE: DEKM

Figure 5: **FMNIST:** Reconstruction loss and clustering quality (1-SC) for all hyperparameter configurations for DCEC, **DC1AM** and DEKM with CAE and RAE architectures. *Lower is better for both axes*, since we plot 1-SC on the $y$-axis.

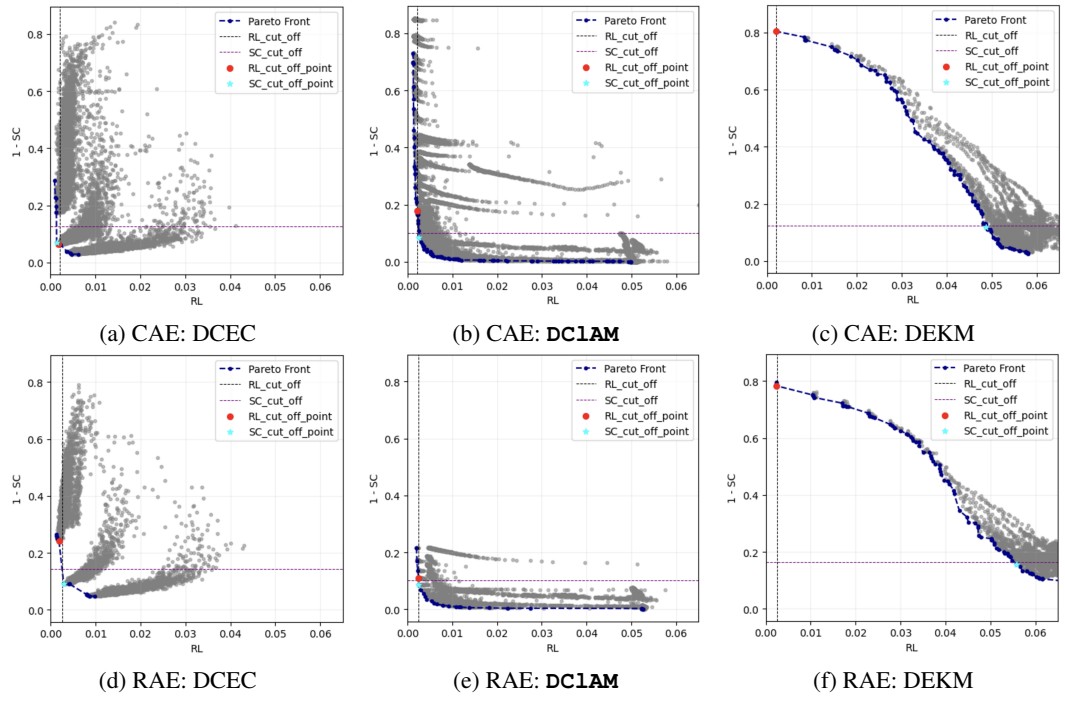

(a) CAE: DCEC  (b) CAE: **DC1AM**  (c) CAE: DEKM

(d) RAE: DCEC  (e) RAE: **DC1AM**  (f) RAE: DEKM

Figure 6: **USPS:** Reconstruction loss and clustering quality (1-SC) for all hyperparameter configurations for DCEC, **DC1AM** and DEKM with CAE and RAE architectures. *Lower is better for both axes*.

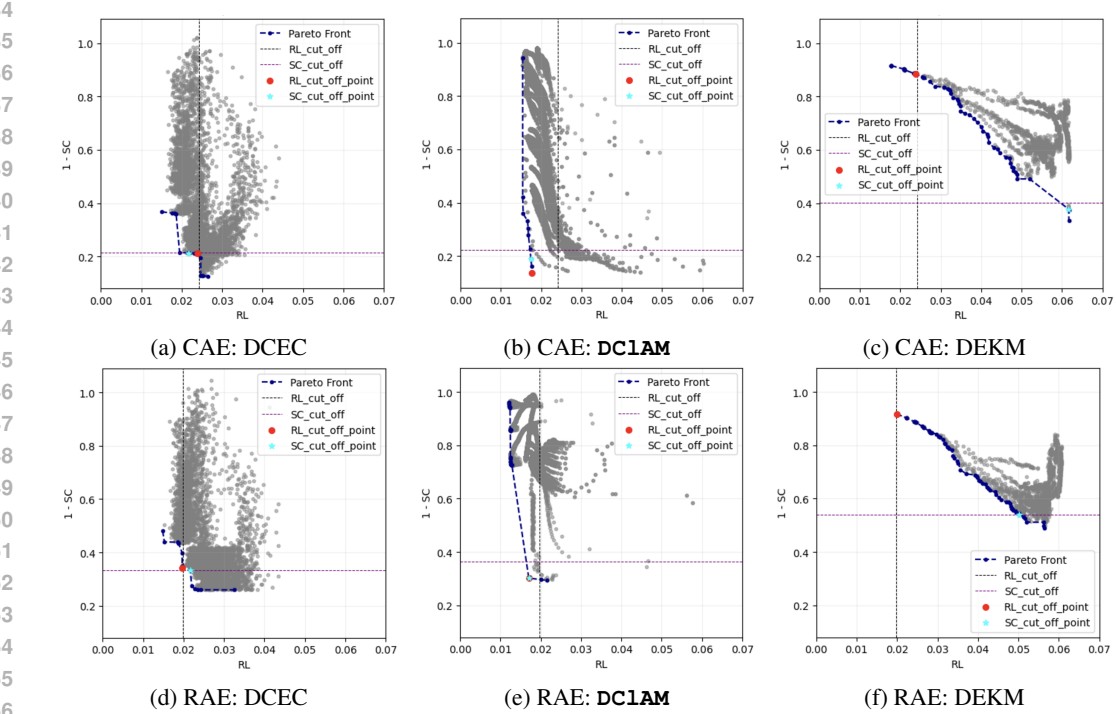

Figure 7: **CIFAR10:** Reconstruction loss and clustering quality (1-SC) for all hyperparameter configurations for DCEC, **DC1AM** and DEKM with CAE and RAE architectures. *Lower is better for both axes.*

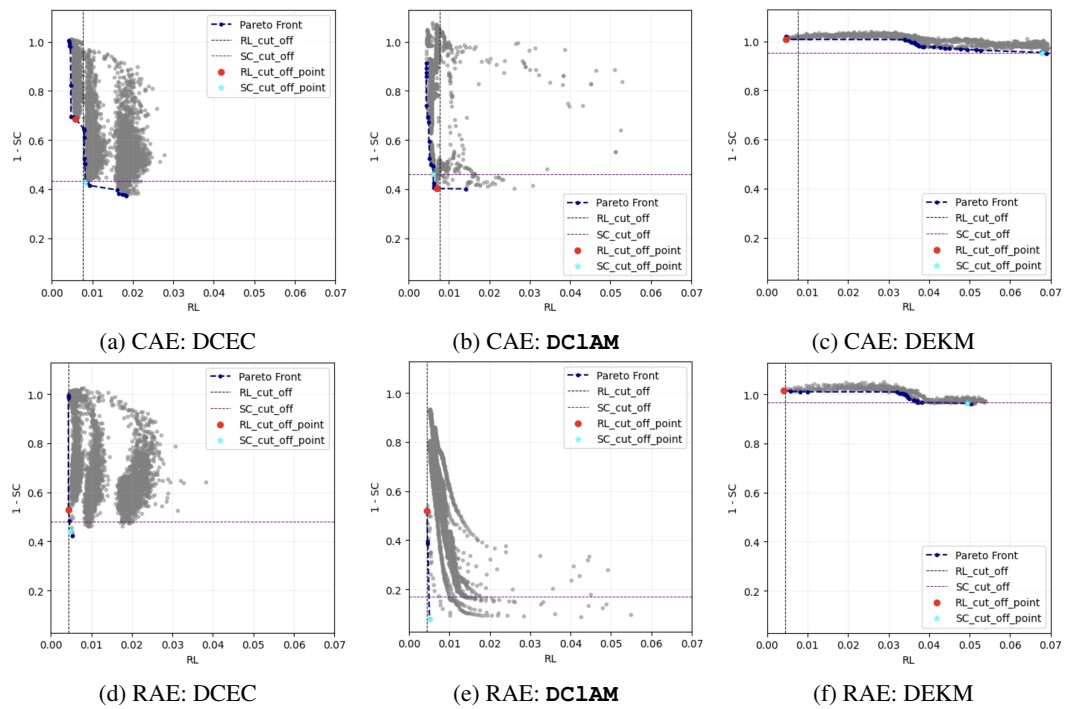

Figure 8: **CIFAR100:** Reconstruction loss and clustering quality (1-SC) for all hyperparameter configurations for DCEC, **DC1AM** and DEKM with CAE and RAE architectures. *Lower is better for both axes.*

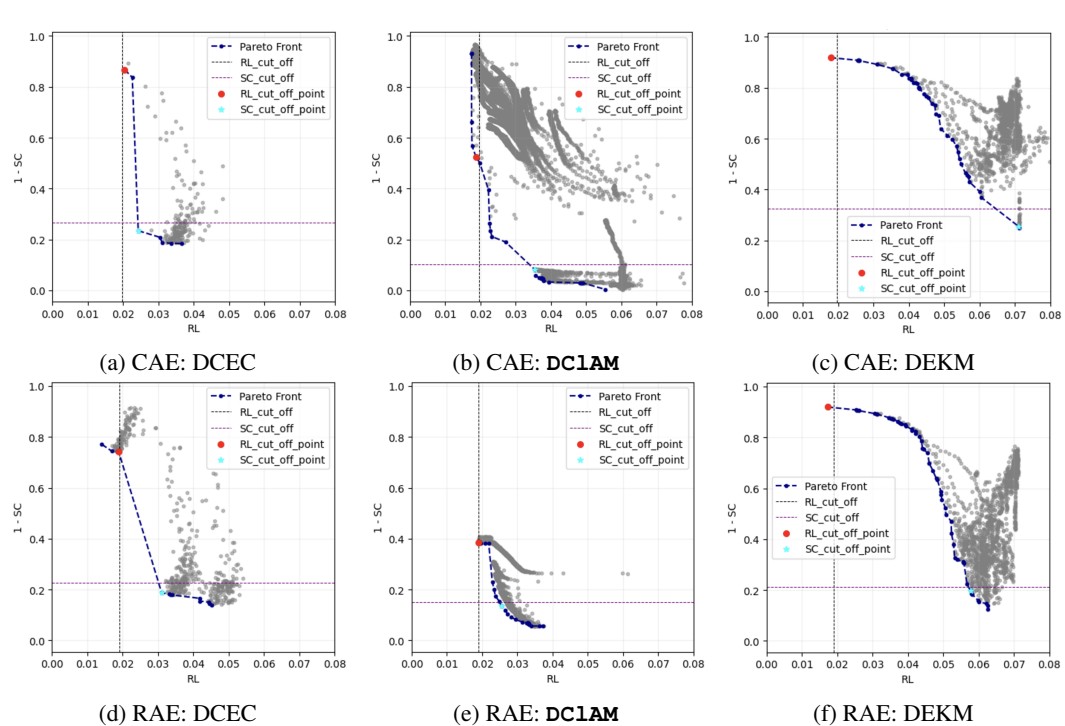

(a) CAE: DCEC      (b) CAE: **DC1AM**      (c) CAE: DEKM

(d) RAE: DCEC      (e) RAE: **DC1AM**      (f) RAE: DEKM

Figure 9: **STL10:** Reconstruction loss and clustering quality (1-SC) for all hyperparameter configurations for DCEC, **DC1AM** and DEKM with CAE and RAE architectures. *Lower is better for both axes*.

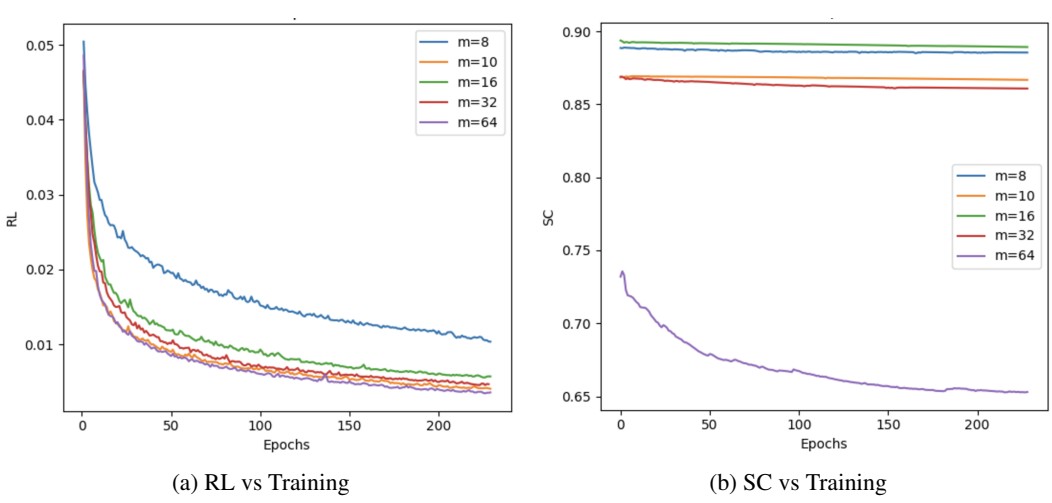

(a) RL vs Training                   (b) SC vs Training

Figure 10: Reconstruction loss (RL) and clustering quality (SC) for varying latent dimension ($m$) for USPS ($T$ is set to 10).

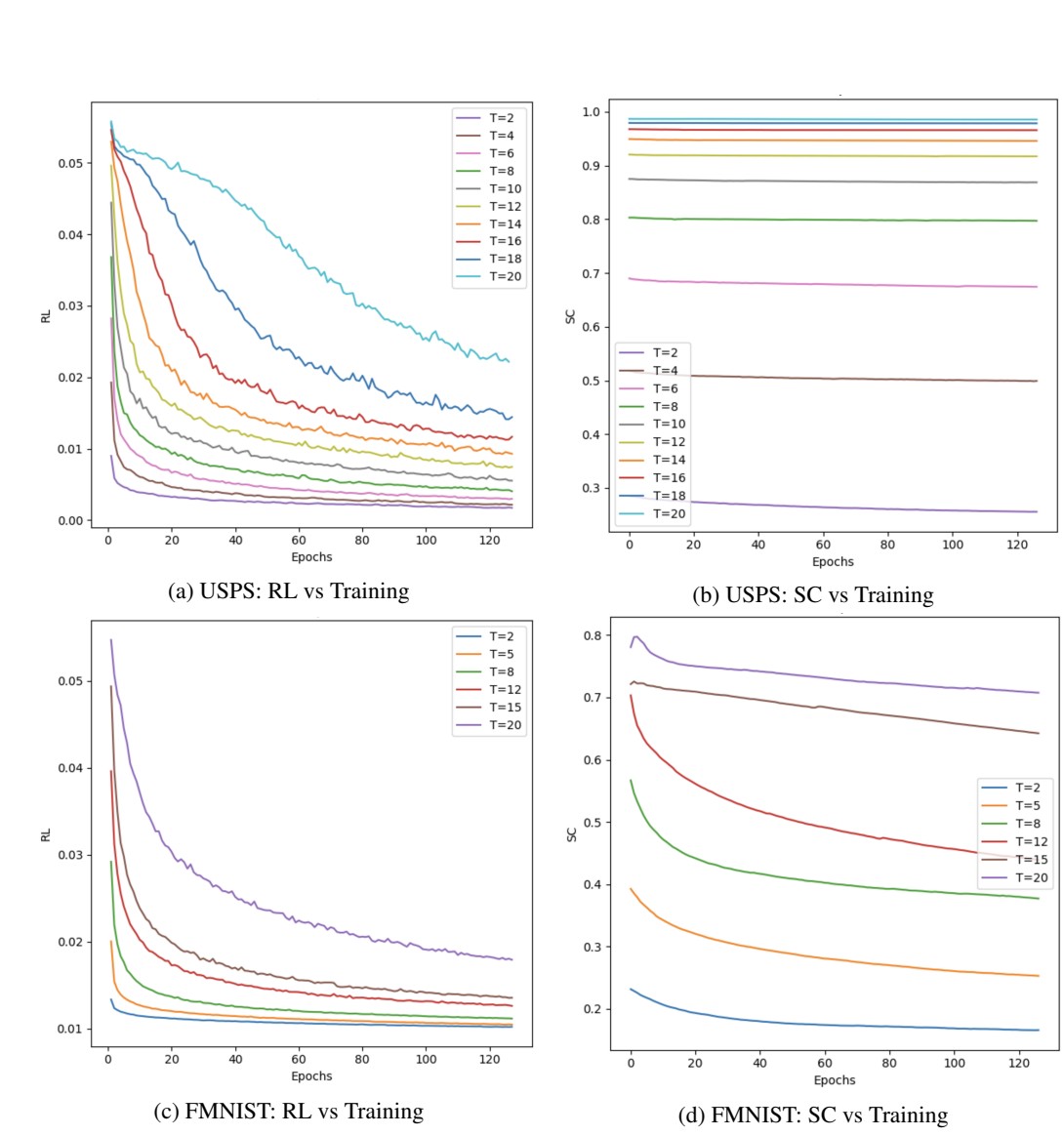

(a) USPS: RL vs Training

(b) USPS: SC vs Training

(c) FMNIST: RL vs Training

(d) FMNIST: SC vs Training

Figure 11: Reconstruction loss (RL) and clustering quality (SC) for varying number of steps (T) for USPS and FMNIST. From the figure, we can see that with increasing T, SC improves while RL worsens.

