# OpenReview forum: "Deep Clustering with Associative Memories"
_ICLR.cc/2025/Conference — Submitted to ICLR 2025_

### Official Review · Reviewer_Z3Ff · 2024-11-03

**Soundness:** 3
**Presentation:** 2
**Contribution:** 2
**Rating:** 6
**Confidence:** 3

**Summary:**

This paper proposes DClAM, a deep clustering method that integrates the associate memory idea of ClAM into the deep learning pipeline. Their method involves fine-tuning an autoencoder model by taking multiple associative memory (AM) steps on encoded embeddings before decoding, which enables joint optimization of representation learning and clustering. The paper shows that DClAM outperforms existing deep clustering methods on eight datasets spanning image and text with multiple architectures.

**Strengths:**

- I think the core idea of integrating AMs into the (deep) autoencoder pipeline via a differentiable loss is novel and interesting. The method is agnostic to architectural choices on the encoder and decoder.
- The paper comprehensively evaluates on diverse datasets, comparing against a range of clustering and deep clustering baselines.

**Weaknesses:**

- There's a large body of work on vector-quantized VAEs [1], which seems related to DClAM in that they're both learning an autoencoder with a latent space with a discrete structure. One obvious difference is that this model isn't a VAE (doesn't consider a prior over latents). Still, I think the learned centers play a similar role to the learned vectors in a VQ-VAE; their objective is essentially yours, with only one AM step. I think the paper should discuss the connection in more detail.
- The paper could benefit from more detailed ablation studies on key hyperparameters. I think reporting sensitivity to (1) the number of AM steps and (2) the number of clusters would give a better sense of the method's tuning requirements. Also, what is the number of AM steps you use? I can't find it in Table 5.
- What does the "restart" hyperparameter mean? Does it mean you ran the entire pipeline with five random seeds and reported the best performance?
- How does this method compare to other deep clustering methods in terms of computational efficiency? Mainly, I'd like to better understand how computationally expensive the AM steps are.

[1] Van Den Oord, Aaron, and Oriol Vinyals. "Neural discrete representation learning." Advances in neural information processing systems 30 (2017).

Minor
- I think it's generally good to have the proposed method's explanation (sec 4) self-contained. A core component (eq 7) is explained separately in section 3; you might consider briefly re-stating this operator in section 4.

**Questions:**

Please see questions in weaknesses section above.

---

> ### Author Response · Authors · 2024-11-23
> **Response**
>
> We want to thank Reviewer Z3Ff for their support of our paper, highlighting how our method is novel and interesting. We respond to specific comments in the following:
>
> > There's a large body of work on vector-quantized VAEs [1], which seems related to DClAM in that they're both learning an autoencoder with a latent space with a discrete structure. One obvious difference is that this model isn't a VAE (doesn't consider a prior over latents). Still, I think the learned centers play a similar role to the learned vectors in a VQ-VAE; their objective is essentially yours, with only one AM step. I think the paper should discuss the connection in more detail.
>
> We thank the reviewer for this insightful comment. There is indeed a connection between $\texttt{DClAM}$ and VQ-VAE in that both of these models learn distinct embedding vectors -- K embedding vectors as discrete latent space for VQ-VAE,  and k memories for $\texttt{DClAM}$. However, there are fundamental differences between them. First, in VQ-VAE, every data point is mapped to one of the stored K (typically 512; and much larger than the number of clusters k) discrete embedding vectors (dictionary mapping), and then this mapped embedding vector is passed through the decoder to construct the original data point to calculate reconstruction loss. In contrast, in $\texttt{DClAM}$, the original data point is not dictionary mapped to one of the stored memories, but rather it moves towards the closest memories after $T$ steps of AM recursion and then it is passed through the decoder. Second, in VQ-VAE, it is not possible to train the original architecture with standard backpropagation as the gradient would not follow through the argmin operation. To overcome this issue, they try to approximate the gradient using straight-through estimator approach (i.e., copying it from the decoder input to encoder output), which loses the ability to minimize the expected loss function. In contrast, in $\texttt{DClAM}$ the AM steps are fully differentiable and can optimize the loss function with full control. Third, VQ-VAE consists of three loss terms (as it loses control of the end-to-end differentiability) whereas $\texttt{DClAM}$ has only one unified loss term. Finally, being a generative model, VQ-VAE focuses on the generation of new data points, whereas $\texttt{DClAM}$ focuses on learning a clustering-friendly autoencoder. As suggested by the reviewer, we will add this discussion in the related work section in our revision.
>
> [1] Van Den Oord, Aaron, and Oriol Vinyals. "Neural discrete representation learning." Advances in neural information processing systems 30 (2017).
>
> > The paper could benefit from more detailed ablation studies on key hyperparameters. I think reporting sensitivity to (1) the number of AM steps and (2) the number of clusters would give a better sense of the method's tuning requirements. Also, what is the number of AM steps you use? I can't find it in Table 5.
>
> We thank the reviewer for this valuable feedback. For the number of AM steps, they are reported in Table 6 across all the datasets and architectures and are usually between 10-15. We plan to include ablation results on varying the AM steps in our revision.
>
> Regarding the number of clusters, it is not a hyperparameter. The number of clusters is chosen as the number of true classes as per dataset. We mentioned it in Table 5 in Appendix (line 770), however, we will certainly include this in the main text to remove the confusion.
>
> > What does the "restart" hyperparameter mean? Does it mean you ran the entire pipeline with five random seeds and reported the best performance?
>
> Yes, the reviewer is correct.
>
> > How does this method compare to other deep clustering methods in terms of computational efficiency? Mainly, I'd like to better understand how computationally expensive the AM steps are.
>
> $\texttt{DClAM}$ (Algorithm 1) has a complexity of $O(dkT N|S|)$, where $d$ is the latent dimension, $k$ is the number of clusters, $T$ is the number of AM steps, $N$ is the number of epochs and $|S|$ is the size of the dataset. Thus, the runtime complexity is linear in the number of AM steps $T$, and it is usually small (10-15 in Table 6 in Appendix).
>
> >  I think it's generally good to have the proposed method's explanation (sec 4) self-contained. A core component (eq 7) is explained separately in section 3; you might consider briefly re-stating this operator in section 4.
>
> Thanks for the suggestion. We will incorporate this in our revision.

---

> ### Author Response · Authors · 2024-11-30
>
> Dear Reviewer Z3Ff,
>
> We would like to once again thank you for taking the time to review our work and your help in improving it. We have addressed all your concerns and included the following in the revised version of our paper.
>
> 1. We have included a discussion of VQ-VAEs in the related work section.
> 2. We have also included ablation results on the effect of varying the number of AM steps $T$ in Appendix C.4 and Figure 11, for both USPS and FMNIST datasets.
>
> We would greatly appreciate it if you could consider increasing your score for our submission.

---

### Official Review · Reviewer_ZsFx · 2024-11-03

**Soundness:** 2
**Presentation:** 3
**Contribution:** 2
**Rating:** 3
**Confidence:** 4

**Summary:**

This study presents DCLAM a deep clustering algorithm that internally uses Clustering with Associative Memory (CLAM) algorithm. The proposed method uses encoder-decoder architecture which is optimized by an associative memory-inspired loss function.  Finally, the method is evaluated in comparison with existing methods using silhouette coefficient (SC) and reconstruction loss over image and text datasets.

**Strengths:**

The research integrates autoencoder and pretraining while optimizing the data in the latent space using the CLAM algorithm. Which is a good extension to CLAM.
The approach tries to maintain a minimal reconstruction loss during the process of finding clusters.
The paper describes the technical details well which make it reproducible.

**Weaknesses:**

The evaluation process only included SC although the ground truth labels are available for the used datasets. The SC alone can’t be enough especially with its underlying assumptions about the cluster’s distributions.
The reported SCs for all the experiments are constrained by 10% range of change in the reconstruction loss. This is limiting the space for clustering improvement during training. Which is not fair for clustering algorithms, especially those that don’t use the decoder anymore in the clustering process.
It is mentioned in the paper that “applying CLAM in a latent space learned by a pretrained autoencoder is not an effective strategy”. However, the results reported don’t test this hypothesis.
The proposed method requires a choice of three learning rates which are difficult to select over different applications.
The belief mentioned in the evaluation section of using unsupervised metrics contradicts with using the information of number of clusters which is unknown in many unsupervised applications.

**Questions:**

Describing the method to be agnostic to architecture can be easily misunderstood. Because architecture is important according to the type of dataset. What could be deduced is that the method itself is flexible and can be integrated with different autoencoders architectures.
Evaluating the proposed method using ground truth-dependent metrics would make the results more reliable. Mitigating data leakage between training and evaluation can still be done without completely ignoring the ground truth information.
The paper lacks a solid justification why balancing the SC and reconstruction loss is important.

---

> ### Author Response · Authors · 2024-11-23
> **Response (Part 1)**
>
> We thank reviewer ZsFx for their thoughtful comments and questions. Below, we aim to address each of the comments provided. We trust that our responses will assist the reviewer in reassessing their evaluation of our paper.
>
> > The evaluation process only included SC although the ground truth labels are available for the used datasets. The SC alone can’t be enough especially with its underlying assumptions about the cluster’s distributions. The reported SCs for all the experiments are constrained by 10\% range of change in the reconstruction loss. This is limiting the space for clustering improvement during training. Which is not fair for clustering algorithms, especially those that don’t use the decoder anymore in the clustering process. It is mentioned in the paper that “applying CLAM in a latent space learned by a pretrained autoencoder is not an effective strategy”. However, the results reported don’t test this hypothesis. The proposed method requires a choice of three learning rates which are difficult to select over different applications. The belief mentioned in the evaluation section of using unsupervised metrics contradicts with using the information of number of clusters which is unknown in many unsupervised applications.
>
> We thank the reviewer for valuable feedback. We address each point in detail below.
>
> 1) We want to mention at the outset that we also evaluated the methods on the Normalized Mutual Information (NMI) metric, which measures how well the ground truth labels correspond to the clusters. We presented the best NMI (and other associated metrics) across all datasets and methods in Table 10 in the Appendix (and also mentioned this in line 414-416 in the main text). Table 10 highlights that $\texttt{DClAM}$ exhibits strong performance not only in terms of unsupervised SC and RL, but also when compared to the ground truth labels via NMI. In fact, for NMI, $\texttt{DClAM}$ has the best value in 5 out of the 8 datasets.
>
>
> 2) Regarding the comment about fairness of reporting the best SC within 10\% of the RL, please note that in general there are two main approaches while clustering in latent space.
> The first option is to pretrain the autoencoder (AE) to minimize the reconstruction loss (RL), and then fix this latent space.  Next, apply some clustering scheme to group the points. In this case, we do not have to consider reconstruction loss while clustering since the AE is fixed.
> However, as we demonstrate experimentally, keeping the autoencoder fixed does not lead to the best clustering. This is shown in the baseline comparison in Tables 2 and 3 that employ a CAE autoencoder, followed by K-means or ClAM in latent space; $\texttt{DClAM}$ outperforms all these baselines in terms of clustering quality; and in several cases it also improves on the RL compared to the fixed autoencoder!
>
> Specifically, regarding ClAM in latent space, we did test this hypothesis and reported all the results in Tables 2, 3, 8, 9 \& 10. We evaluate ClAM both in the ambient space (denoted as NAE for No AE) and in the latent space obtained through a pretrained Convolutional Autoencoder (CAE). And we found that running ClAM in the (pretrained) latent space is not an effective clustering strategy while $\texttt{DClAM}$ outperforms ClAM with a very high margin utilizing its intrinsic clustering-friendly architecture.
>
> Returning to the second option for clustering in latent space, instead of fixing the encoding, we can fine-tune the autoencoder (i.e., both the encoder and decoder) so that is modifies the latent space along with the task of finding clusters. Here, if we do not consider reconstruction loss, then it means that there is effectively no decoder, since there is no RL constraint, and this can lead to trivial clusterings. As an extreme example, the encoder can map each point to its corresponding center, which would lead to a trivial clustering, but one that has a sum of squared errors loss as zero (or if we consider Silhouette Coefficient (SC), this trivial clustering will have a SC value of 1). In other words, if we do want to fine-tune both the encoder and decoder, to allow more flexibility to learn a better clustering-guided latent space, then considering RL is a must; it acts as a regularization constraint on the latent space. The key insight and contribution of $\texttt{DClAM}$ is that we seamlessly combine the clustering and RL objectives into one expression that tackles the task of clustering-guided latent representations, whereas previous deep clustering methods considered these separately.

---

> ### Author Response · Authors · 2024-11-23
> **Response (Part 2)**
>
> > Continuing from part 1 ...
>
> For fairness sake, we argue the opposite. Since it is not immediately clear whether SC or RL should have more weight, we report in Table 2 the best SC value obtained when ensuring that RL does not degrade by more than 10\% of the initial pretrained AE. We also report in Table 3, the best RL for the methods when SC is within 10\% of the best SC obtained by the method. Table 8 and 9 give more detailed results for these two cases, respectively. What we observe is that in either of these two approaches $\texttt{DClAM}$ has the best results for the SC, and best RL in most of the cases.
>
> 3) We appreciate the reviewer's concern regarding the three learning rates of $\texttt{DClAM}$. We would like to emphasize that there are two main hyperparameters for $\texttt{DClAM}$, namely inverse temperature ($\beta$), and number of steps of AM ($T$) (line 760-761). The other hyperparameters listed in Table 5 in Appendix (762-221), such as batch size, learning rate, patience, and so on are very common for almost all deep clustering/learning schemes. Furthermore, it is clear from Table 6 in the appendix that most of these hyperparameter values can be quite stable across the datasets. For example, there is little variation in the encoder, decoder or AM learning rates. The same is true for batch size, and the number of AM steps ($T$) is usually 10-15. The only parameter that requires some tuning is $\beta$, whose optimal value ranges from 0.00015 to 10. In general, the convolutional autoencoder (CAE) is less sensitive to $\beta$, whereas resnet autoencoder (RAE) is more sensitive, with typically smaller values of $\beta$ for larger latent dimensions ($m$), which corresponds to the number of true classes in the dataset ($k$).
>
> 4) Finally, we agree with the reviewer that in the pure unsupervised process, the number of clusters is unknown. However, we are using this information only for how many clusters we want to find, which is also the user input required for virtually all clustering methods. However, in future work, we do plan to explore how we can leverage our approach to automatically find the true number of clusters (we also mentioned this limitation in line 485).
>
> > Describing the method to be agnostic to architecture can be easily misunderstood. Because architecture is important according to the type of dataset. What could be deduced is that the method itself is flexible and can be integrated with different autoencoders architectures. Evaluating the proposed method using ground truth-dependent metrics would make the results more reliable. Mitigating data leakage between training and evaluation can still be done without completely ignoring the ground truth information. The paper lacks a solid justification why balancing the SC and reconstruction loss is important.
>
> Thanks for suggesting the point about ``agnostic''. We agree that is better to state that our approach can flexibly be combined with different AE architectures. We will rephrase this statement.
>
> Regarding the ground-truth-dependent metrics, we responded to this in point 2 above. If we employ autoencoder pretraining, then we can optimize for the clustering quality (such as SC) while ensuring that the reconstruction loss is within some margin (say 10\%) of the reconstruction loss of the pretrained autoencoder. In our response 2) above we provide the justification why it is critical to balance the clustering quality with reconstruction loss when allowing both the encoder and decoder to evolve while clustering.
>
> Finally, we believe that the hyperparameters for clustering and clustering evaluation should ideally be based on unsupervised metrics, without utilizing ground-truth label information. Nevertheless, we do compare the methods on NMI too and show that $\texttt{DClAM}$ retains its superiority even on this supervised metric.

---

> ### Author Response · Authors · 2024-11-29
>
> Dear Reviewer ZsFx,
>
> We would like to once again thank you for taking the time to review our work and for your help in improving it. We have addressed all your concerns in the revised version of our paper. We would greatly appreciate it if you could consider increasing your score for our submission.

---

### Official Review · Reviewer_AgR2 · 2024-11-04

**Soundness:** 3
**Presentation:** 3
**Contribution:** 2
**Rating:** 6
**Confidence:** 2

**Summary:**

The paper applies ClAM (Saha et al. 2023) to various neural architectures (CNN, MLP) and achieves competitive performances on various text and image clustering benchmarks. The main metric of concern is the Silhouette Coefficient (SC), commonly used in clustering literature for unsupervised clustering quality.

**Strengths:**

- Paper is well written and easy to follow
- Very clear motivation, theoretical analysis appear to be correct, core algorithm clearly explained, and experiments are presented thoroughly

**Weaknesses:**

- The work builds up on ClAM (Saha et al., 2023), which may limit its novelty.
- Arguably, since the work is concerned with deep clustering, baseline methods should include traditionally metric learning-based approaches.
- In contemporary literature standards, both the neural network trained in the work and the dataset are small. It is unclear whether the proposed method would generalize to more realistic network sizes and datasets.

**Questions:**

- It seems like the entire (vision) transformer literature is missing from the analysis. This is not a critical concern, but it would be interesting to evaluate DClAM's performance on transformers across image and text tasks.

---

> ### Author Response · Authors · 2024-11-23
> **Response (Part 1)**
>
> We thank reviewer AgR2 for their review and kind words of support for our paper, acknowledging its "well-written" and "very clear motivation". We address their specific comments in the sections below and wish the reviewer to reconsider their evaluation of our paper towards acceptance.
>
> > The work builds up on ClAM (Saha et al., 2023), which may limit its novelty.
>
> We would like to emphasize that there are fundamental differences between ClAM and $\texttt{DClAM}$. The former proposes a differential clustering scheme in the input space utilizing associative memories (AM), whereas our $\texttt{DClAM}$ approach focuses on clustering in latent space, utilizing AM to find good clusters and yet retaining good reconstruction (which reflects the quality of the latent representations). Achieving the joint goal of clustering-guided latent representation is a novel task.
>
> Furthermore, we did elucidate the key differences between them in our submission. The discussion in lines 293-300 points out that ClAM, AM is utilized as a differentiable argmin solver for the $k$-means objective, whereas in $\texttt{DClAM}$, which involves representation learning, AM recursion actually has a more elaborate effect. The AM augmented encoder ($A_{\mathbf{\rho}}^T \circ \mathbf{e}$) explicitly creates basins of attraction in the latent space, and moves/pushes the latent representations of the points into these basins, thereby explicitly inducing a clustered data distribution in the latent space. While the encoder is moving points into basins of attraction, the $\texttt{DClAM}$ loss tries to minimize the information loss in the latent representations by having the decoder reconstruct these relocated latent representations.
>
> The discussion in lines 280-290 lists the various advantages of $\texttt{DClAM}$, namely (i) First, it does not involve any balancing hyperparameter $\gamma$ since the loss involves all parameters in a single term in the per-sample loss $\bar \ell(x, \mathbf{e}, \mathbf{d}, {\mathbf{\rho}})$. (ii) Second, the updates for all the parameters in the pipeline are more explicitly tied together with the $\mathbf{d} \circ A_{\mathbf{\rho}}^T \circ \mathbf{e}$ composition in the $\mathbf{d}( A_{\mathbf{\rho}}^T( \mathbf{e}( x ) ) )$ term. This ties the representation learning and clustering objectives more intricately. (iii) Third, it continues to have all the advantages of
> traditional deep clustering, being end-to-end differentiable since all operators in the above composition are differentiable, and performing a discrete cluster center assignment with $T$ recursions of the attractor dynamics operator $A_{\mathbf{\rho}}$. (iv) Forth, this deep clustering can be combined with different auto-encoder architectures -- we can select a problem dependent encoder and decoder (for example, convolutional or residual networks for images or fully-connected feed-forward networks for text or tabular data).
> (v) Fifth, it does not involve any additional entropy regularization based hyperparameters as with existing deep clustering algorithms.
>
> To highlight these key contributions and differences, we will certainly restructure our introduction section emphasizing these unique aspects of $\texttt{DClAM}$.
>
> > Arguably, since the work is concerned with deep clustering, baseline methods should include traditionally metric learning-based approaches.
>
> We want to mention that in our study, we prioritized baseline methods that align closely with the primary focus of our work, namely deep clustering models that jointly learn representations and cluster assignments. Metric learning
> refers to the task of learning the ``similarity'' between pairs of points. In an unsupervised setting the complexity of this task is at least $O(|S|^2)$ where $|S|$ is the dataset size (number of points). Therefore, metric learning is typically performed either in a supervised setting where the label is known and we learn a distance metric that maximizes similarity between points in the same class and minimize it between points across classes. Alternatively, it can be done in a weakly supervised setting, where pairs of close and far away points are given as positive and negative sets, with the goal of learning a distance metric that puts positive pairs close together and negative pairs far away.
>
> As such metric learning task is not directly comparable to the objectives of unsupervised deep clustering, with the aim of learning a latent representation and clustering at the same time.
> If the reviewer has any specific metric learning approach in mind, we will be happy to discuss that.

---

> ### Author Response · Authors · 2024-11-23
> **Response (Part 2)**
>
> >  In contemporary literature standards, both the neural network trained in the work and the dataset are small. It is unclear whether the proposed method would generalize to more realistic network sizes and datasets.
>
> We have experimented with a diverse set of standard benchmark datasets ranging from 2000 to 60000 data points, and from 10 to 200 clusters, that cover most of the commonly used datasets in deep clustering literature.
>
> As such, our method should scale to much larger datasets, since the Train subroutine of $\texttt{DClAM}$ in algorithm 1 has a linear complexity of $O(dkT N|S|)$, where $d$ is the latent dimension, $k$ is the number of clusters, $T$ is the number of AM steps, $N$ is the number of epochs and $|S|$ is the dataset size. Also, $\texttt{DClAM}$ leverages SGD and works on batches of training data, so we anticipate that there are no barriers to scaling to larger datasets. Another thing we want to mention regarding the network size is that $\texttt{DClAM}$ works seamless with any choice of encoder and decoder (for example, convolutional or residual networks for images or fully-connected feed-forward networks for text or tabular data, and so on).
>
> > It seems like the entire (vision) transformer literature is missing from the analysis. This is not a critical concern, but it would be interesting to evaluate DClAM's performance on transformers across image and text tasks.
>
> We thank the reviewer for suggesting this. As we note, our framework can easily utilize any encoder/decoder framework. In the future we will also explore transformer based AE approaches.

---

> ### Author Response · Authors · 2024-11-29
>
> Dear Reviewer AgR2,
>
> We would like to once again thank you for taking the time to review our work and your help in improving it. We have addressed all your concerns and included the fundamental differences between DClAM and ClAM, in the revised introduction and related works section of our paper. We have added the relation to metric learning too.
>
> We would greatly appreciate it if you could consider increasing your score for our submission.

---

> > ### Comment · Reviewer_AgR2 · 2024-12-02
> >
> > Thank you; I'm satisfied with the updated version and have adjusted my score accordingly.

---

> > > ### Author Response · Authors · 2024-12-02
> > > **Thank you!**
> > >
> > > Thank you very much! We are very happy to hear that you are satisfied with our responses. Given that the decision-making process will soon be underway, if you believe  that our paper is worthy of acceptance, we kindly ask considering a further increase of the score to a "full accept" (8). We remain available to resolve any outstanding issues that would bump our paper from a borderline to an accept in the remaining time dedicated for discussion.

---

### Official Review · Reviewer_whZj · 2024-11-06

**Soundness:** 3
**Presentation:** 3
**Contribution:** 3
**Rating:** 6
**Confidence:** 3

**Summary:**

This paper presents DClAM, a deep clustering method that builds on the previous ClAM approach by integrating it with deep clustering techniques, specifically through a deep autoencoder (AE). DClAM is architecture-agnostic, demonstrating strong performance across different autoencoder designs and dataset types. It achieves high clustering quality and low reconstruction loss in both ambient and latent spaces, surpassing previous methods.

**Strengths:**

The paper demonstrates superior results across various benchmarks, highlighting the robustness of the proposed approach. Additionally, the authors conduct experiments across diverse domains and architectural types, effectively showcasing the generalizability of their method. The preliminary section provides a thorough overview of foundational concepts, enhancing the accessibility of the paper. Furthermore, the appendix includes detailed hyperparameter information, essential for validating experiments in this hyperparameter-sensitive area and supporting the reproducibility of the study.

**Weaknesses:**

First and foremost, this paper resembles the CLAM paper strongly, particularly in the introduction section, which feels almost identical. Given that this work relies heavily on CLAM, it is essential to revise and reframe the introduction to clearly distinguish this approach as more than a direct application of CLAM with an autoencoder. Establishing this work's unique contributions will help clarify its originality and value.

While the authors claim that the method does not require $\gamma$ tuning, it does not appear to resolve the underlying parameter sensitivity issue. In fact, it introduces additional hyperparameters that need careful adjustment. The chosen values suggest that previous tuning decisions are not reusable across different datasets and models, requiring high tuning complexity. This is particularly challenging, given that model training for reconstruction must occur simultaneously with the tuning process.

 An essential missing experiment is an ablation study on the latent dimension \( m \). Evaluating this parameter, even for one of the models used, would provide valuable insights into the impact of latent dimensionality on performance.

**Questions:**

How is the number of centers (clusters) chosen? It’s unclear whether this is a fixed number or if it corresponds to the number of classes in the dataset.

Why is reconstruction considered significant in this context? It seems that the assumption of minimal information loss in the latent space is a prerequisite for effective clustering. To my understanding, a low reconstruction loss simply indicates that good clustering may be achievable, rather than serving as a goal in itself.

---

> ### Author Response · Authors · 2024-11-22
> **Response (Part 1)**
>
> We thank reviewer whZj for their thoughtful comments and questions. In what follows, we try to address each of the comments. We hope that these help the reviewer reconsider their evaluation of our paper.
>
> > First and foremost, this paper resembles the CLAM paper strongly, particularly in the introduction section, which feels almost identical. Given that this work relies heavily on CLAM, it is essential to revise and reframe the introduction to clearly distinguish this approach as more than a direct application of CLAM with an autoencoder. Establishing this work's unique contributions will help clarify its originality and value.
>
> We want to thank the reviewer for this valuable feedback.
>
> There are fundamental differences between ClAM and $\texttt{DClAM}$. The former proposes a differential clustering scheme in the input space utilizing associative memories (AM), whereas our $\texttt{DClAM}$ approach focuses on clustering in latent space, utilizing AM to find good clusters and yet retaining good reconstruction (which reflects the quality of the latent representations). In fact, we do mention the key differences in detail in our submission, in lines 293-300 and 280-290.
>
> The discussion in lines 293-300 points out that in ClAM, AM is utilized as a differentiable argmin solver for the $k$-means objective, whereas in $\texttt{DClAM}$, which involves representation learning, AM recursion actually has a more elaborate effect. The AM augmented encoder ($A_{\mathbf{\rho}}^T \circ \mathbf{e}$) explicitly creates basins of attraction in the latent space, and moves/pushes the latent representations of the points into these basins, thereby explicitly inducing a clustered data distribution in the latent space. While the encoder is moving points into basins of attraction, the $\texttt{DClAM}$ loss tries to minimize the information loss in the latent representations by having the decoder reconstruct these relocated latent representations.
>
> The discussion in lines 280-290 lists the various advantages of $\texttt{DClAM}$, namely (i) First, it does not involve any balancing hyperparameter $\gamma$ since the loss involves all parameters in a single term in the per-sample loss $\bar \ell(x, \mathbf{e}, \mathbf{d}, {\mathbf{\rho}})$. (ii) Second, the updates for all the parameters in the pipeline are more explicitly tied together with the $\mathbf{d} \circ A_{\mathbf{\rho}}^T \circ \mathbf{e}$ composition in the $\mathbf{d}( A_{\mathbf{\rho}}^T( \mathbf{e}( x ) ) )$ term. This ties the representation learning and clustering objectives more intricately. (iii) Third, it continues to have all the advantages of
> traditional deep clustering, being end-to-end differentiable since all operators in the above composition are differentiable, and performing a discrete cluster center assignment with $T$ recursions of the attractor dynamics operator $A_{\mathbf{\rho}}$. (iv) Forth, this deep clustering can be combined with different auto-encoder architectures -- we can select a problem dependent encoder and decoder (for example, convolutional or residual networks for images or fully-connected feed-forward networks for text or tabular data).
> (v) Fifth, it does not involve any additional entropy regularization based hyperparameters as with existing deep clustering algorithms.
>
> However, to highlight these key contributions and differences, we will certainly restructure our introduction section emphasizing these unique aspects of $\texttt{DClAM}$.

---

> ### Author Response · Authors · 2024-11-23
> **Response (Part 2)**
>
> > While the authors claim that the method does not require $\gamma$
>  tuning, it does not appear to resolve the underlying parameter sensitivity issue. In fact, it introduces additional hyperparameters that need careful adjustment. The chosen values suggest that previous tuning decisions are not reusable across different datasets and models, requiring high tuning complexity. This is particularly challenging, given that model training for reconstruction must occur simultaneously with the tuning process.
>
> We appreciate the reviewer's concern regarding the hyperparameter sensitivity of $\texttt{DClAM}$. We would like to emphasize that there are two main  hyperparameters for $\texttt{DClAM}$, namely inverse temperature ($\beta$), and number of steps of AM ($T$) (line 760-761). The other hyperparameters listed in Table 5 in Appendix (762-221), such as batch size, learning rate, patience, and so on are very common for almost all deep clustering/learning schemes.
>
> Furthermore, it is clear from Table 6 in the appendix that most of these hyperparameter values can be quite stable across the datasets. For example, there is little variation in the encoder, decoder or AM learning rates. The same is true for batch size, and the number of AM steps ($T$) is usually 10-15. The only parameter that requires some tuning is $\beta$, whose optimal value ranges from 0.00015 to 10. In general, the convolutional autoencoder (CAE) is less sensitive to $\beta$, whereas resnet autoencoder (RAE) is more sensitive, with typically smaller values of $\beta$ for larger latent dimensions ($m$), which corresponds to the number of true classes in the dataset ($k$).
> It is worth noting that while $\texttt{DClAM}$ requires tuning $\beta$, it is a completely new strategy for deep clustering using associative memories that outperforms all the related baseline schemes.
>
> > An essential missing experiment is an ablation study on the latent dimension (m). Evaluating this parameter, even for one of the models used, would provide valuable insights into the impact of latent dimensionality on performance.
>
> It is important to clarify that in $\texttt{DClAM}$ the latent dimensions $m$ is always set as the true number of classes per dataset, i.e., $m=k$. Indeed, most deep clustering schemes in the literature such as DCEC[1], DEC [2], DEKM[3], and EDCWRN [4] either follow this strategy or fix this to a specific number (e.g., 10), since latent representations are not only just good representations of the data points, they also represent the clusters. In $\texttt{DClAM}$, we always set this latent dimension as the number of clusters. By setting this exactly the same as the number of clusters, each latent dimension should ideally represent one specific cluster. If the latent dimensions are larger than the number of clusters, some dimensions might not align with any specific cluster, or multiple dimensions could end up representing the same cluster. This can introduce redundancy and result in a less efficient representation. On the other hand, if the latent dimensions are smaller than the number of clusters, some clusters may not be adequately represented. This forces multiple clusters to share the same dimension, making it challenging for the model to distinguish between them accurately.
>
> We can also consider a spectral argument [5] for setting $m=k$ .
> Given a set of $n$ points (in any representation) from $k$ ground-truth clusters, consider a graph (weighted or
> unweighted and undirected) with each point as a node, and edges between points belonging to the same cluster, and no
> inter-cluster edges. This graph would have $k$ connected components, and the Laplacian $L \in \mathbb{R}^{n \times n}$ of
> this graph will have $k$ zero eigenvalues (for example, see Von Luxburg [5, Proposition 2]). Now consider the first $k$
> eigenvectors $u_1, \ldots, u_k \in \mathbb{R}^n$ forming the columns of the matrix $U \in \mathbb{R}^{n \times k}$. Then
> each row $z_i \in \mathbb{R}^k$ of $U$ can serve as a representation of the point $i$, and the points will be
> well-separated into $k$ clusters in this representation. This is the intuition that forms the basis of various spectral clustering algorithms.

---

> ### Author Response · Authors · 2024-11-23
> **Response (Part 3)**
>
> > Continuing from part 2 ...
>
> The above implies the existence of a $k$-dimensional space where the points (coming from $k$ ground-truth
> clusters) are well-separated into $k$ clusters. Thus, a latent space of $k$ dimensions is necessary to obtain
> well-separated clusters. As Euclidean clustering becomes more challenging with increasing representation dimensionality
> (the representation in which the clustering is happening), the motivation is to keep the latent space dimension as low
> as possible as long as we have enough dimensions to separate the clusters. For this reason, the latent space
> dimensionality is most deep clustering usually matches the desired number of clusters (as long as the Euclidean
> clustering is happening with the latent representations). A higher latent dimensionality will definitely help with the
> reconstruction but can potentially hurt Euclidean clustering; a lower latent dimensionality would not be sufficient to
> obtain $k$ well-separated clusters.
> Fortunately, given the extremely expressive modern deep learning encoder and decoders, we are able to still get quite low reconstruction loss with a $m=k$ dimensional latent space.
>
> [1] Xifeng Guo, Xinwang Liu, En Zhu, and Jianping Yin. Deep clustering with convolutional autoencoders. In International conference on neural information processing, pp. 373–382. Springer, 2017b.
>
> [2] Junyuan Xie, Ross Girshick, and Ali Farhadi. Unsupervised deep embedding for clustering analysis. In International conference on machine learning, pp. 478–487. PMLR, 2016.
>
> [3] Wengang Guo, Kaiyan Lin, and Wei Ye. Deep embedded k-means clustering. In 2021 International
> Conference on Data Mining Workshops (ICDMW), pp. 686–694. IEEE, 2021.
>
> [4] Amin G. Oskouei, Mohammad A. Balafar, and Cina Motamed. Edcwrn: efficient deep clustering with
> the weight of representations and the help of neighbors. Applied Intelligence, 53(5):5845–5867,
> 2023.
>
> [5] Von Luxburg, Ulrike. A tutorial on spectral clustering. Statistics and computing 17 (2007): 395-416.
>
> > How is the number of centers (clusters) chosen? It’s unclear whether this is a fixed number or if it corresponds to the number of classes in the dataset.
>
> We apologize for any confusion. The number of centers (clusters) is not a hyperparameter; they are always chosen as number of true classes in each dataset, i.e., $m=k$. We did mentioned this in Table 5 in Appendix (line 770), but we will reemphasize this in the main text.
>
> > Why is reconstruction considered significant in this context? It seems that the assumption of minimal information loss in the latent space is a prerequisite for effective clustering. To my understanding, a low reconstruction loss simply indicates that good clustering may be achievable, rather than serving as a goal in itself.
>
> We thank the reviewer for this insightful question. We agree that the assumption of minimal reconstruction loss is a prerequisite for a good clustering in latent space.
> However, there are mainly two options while clustering in latent space.
>
> First option is to pretrain the autoencoder (AE) to minimize the reconstruction loss (RL), and then fix this latent space.  Next, apply some clustering scheme to group the points. In this case, we do not have to consider reconstruction loss while clustering since the AE is fixed.
> However, as we demonstrate experimentally, keeping the autoencoder fixed does not lead to the best clusterings. See for example the baseline comparison in Tables 2 and 3 that employs a CAE autoencoder, followed by K-means or ClAM in latent space; $\texttt{DClAM}$ outperforms all these baselines in terms of clustering quality; and in several cases it also improves on the RL compared to the fixed autoencoder!
>
> The second option is to fine-tune the autoencoder (i.e., both the encoder and decoder) so that it modifies the latent space along with the task of finding clusters.
> In fact, if we do not consider reconstruction loss, then it means that there is effectively no decoder, since there is no RL constraint, and this can lead to trivial clusterings. As an extreme example, the encoder can map each point to its corresponding center, which would lead to a trivial clustering, but one that has a sum of squared errors loss as zero (or if we consider Silhouette Coefficient (SC), this trivial clustering will have a SC value of 1).
>
> In other words, if we do want to fine-tune both the encoder and decoder, to allow more flexibility to learn a better ``clustering-guided'' latent space, then considering RL is a must; it acts as a regularization constraint on the latent space. The key insight and contribution of $\texttt{DClAM}$ is that we seamlessly combine the clustering and RL objectives into one expression that tackles the task of clustering-guided latent representations, whereas previous deep clustering methods considered these separately.

---

> ### Comment · Reviewer_whZj · 2024-11-24
>
> Dear Authors,
>
> Thank you for addressing my concerns in your responses. While I have no further questions, I would like to review a revised version of the paper to reconsider my score.

---

> > ### Author Response · Authors · 2024-11-26
> >
> > Dear Reviewer,
> >
> > We have submitted a revised version of our paper, and we cordially request you to take a review of this revised version to reconsider our paper toward acceptance. Thank you for your kind support of our paper.

---

> ### Author Response · Authors · 2024-11-29
>
> Dear Reviewer whZj,
>
> We would like to once again thank you for taking the time to review our work and your help in improving it. We have addressed all your concerns in the revised version of our paper. We would greatly appreciate it if you could consider increasing your score for our submission.

---

> > ### Comment · Reviewer_whZj · 2024-12-02
> >
> > Dear Authors,
> >
> > I am satisfied with the revised version and have adjusted my score accordingly.

---

> ### Author Response · Authors · 2024-12-02
>
> Dear Reviewer whZj,
>
> We sincerely appreciate your support of our paper.

---

### Author Response · Authors · 2024-11-26
**General response (with revised version)**

We sincerely thank all reviewers for their valuable comments, which we
have thoroughly addressed in our responses below, as well as in the revised
manuscript that has been updated on openreview. All changes have been
highlighted in blue in the main text and the Appendix. We summarize our main
changes below.

As requested by reviewer whZj we have revised the introduction and related
work sections, to outline the key differences between DClAM and ClAM and
other deep clustering works. Likewise, we have added the ablation study by
varying the latent dimension $m$ for USPS in Appendix C.4, Fig 10. We have
also clarified that the number of clusters $k$ is not a hyperparameter (end of
sec A.2).

For reviewer AgR2, we note the fundamental differences between DClAM and
ClAM, as now done in the revised introduction and related works section. We
have added the relation to metric learning too.

For reviewer ZsFx, we have added new experiments on the Pareto frontier for
all the hyperparameter configurations for different vision-based datasets in
Appendix C.2, and Figures 5,6,7,8,9. We also show the Pareto optimal
parameters reported for the methods for the best SC and RL, and within the
10% thresholds, as reported in Tables 2,3,4 and Tables 8,9. These results
highlight how we thoroughly optimize the hyperparameters, and how we select
the final Pareto optimal performance values from the Pareto front to be
consistent and fair across all methods. The results clearly indicate that
DClAM offers the best clustering performance in terms of SC, as well as
having low reconstruction loss. It also performs very well on the
supervised NMI metric. In fact, for NMI, DClAM has the best value in 5 out
of the 8 datasets.

Finally, for reviewer Z3Ff, we include a discussion of VQ-VAEs in the related
work section. We also include ablation results on the effect of varying
the number of AM steps $T$ in Appendix C.4 and Figure 11, for both USPS and
FMNIST datasets.

We request the reviewers to reconsider their evaluation of our paper in
light of these revisions.

---

### Author Response · Authors · 2024-12-02

Dear Reviewers, Dear Area Chair,

Thanks a lot for taking the time to read our revised manuscript! We are very happy to hear that reviewers whZj and AgR2 are satisfied with our response and have updated their scores.

There is still one reject score (3) from reviewer ZsFx, and we have not received any response from them regarding our rebuttal and revised manuscript. Since there is less than 24 hours left for us to be able to communicate with reviewers, we kindly ask reviewer ZsFx to consider increasing the numerical score for our submission.

We are thankful to reviewers whZj and AgR2 for raising their scores to 6. With the decision-making process soon underway, if whZj or AgR2 believe our paper is worthy of acceptance, we kindly ask considering a further increase of the score to a "full accept" (8). We remain available to resolve any outstanding issues that would bump our paper from a borderline to an accept in the remaining time dedicated for discussion.

Sincerely, Authors

---

### Author Response · Authors · 2024-12-04
**Summary of Review and Response**

We express our sincere gratitude to the Reviewers and Area Chairs for their dedicated time and constructive feedback. Here is a concise summary of the review and our responses for easy reference.

**Reviewer Acknowledgments:**

Our method, $\texttt{DClAM}$, has been recognized for its **Novelty** and **Superior Results**. Key highlights include:

- **Clear Motivation and Novelty:** $\texttt{DClAM}$ comes up with **very clear motivation** with precise **theoretical analysis** and explanation of the core algorithm [AgR2]. The core idea of integrating AMs into the (deep) autoencoder pipeline via a differentiable loss is **novel** and **interesting** [Z3Ff].

- **Superior Results, Generalizability and Reproducibility:** $\texttt{DClAM}$ achieves superior results across various benchmarks with diverse domains and architectural types, effectively showcases the **generalizability** of the method [whZj]. It describes the technical details well which make it **reproducible** [ZsFx]. Moreover, $\texttt{DClAM}$ is **agnostic** to architectural choices on the encoder and decoder that makes it easy to incorporate with any architecture [Z3Ff].

- **Quality of Writing:** The manuscript is praised as **"well written"** and **"easy to follow"** [AgR2], making it easily **accessible** to the field [whZj].

**Addressing Weaknesses:**

The reviewers raised concerns regarding the clarification of unique contributions of $\texttt{DClAM}$ [whZj, AgR2], hyperparameter tuning [whZj, ZsFx], more ablation studies [whZj, Z3Ff], generalization [AgR2] and evaluation metrics [ZsFx]. Our responses include:

- *Unique contributions of $\texttt{DClAM}$:* We clarified in this [comment](https://openreview.net/forum?id=Awsb8jhEx3&noteId=cs6uLwJvJB) the unique contributions of $\texttt{DClAM}$ and fundamental differences from $\texttt{ClAM}$ and other deep clustering methods.

- *Ablation studies:* Following suggestions from Reviewers whZj and Z3Ff, we have added the ablation studies by varying the **latent dimension $\textbf{m}$** for USPS and varying the number of **AM steps $\textbf{T}$** for both USPS and FMNIST datasets.

- *HP tuning and generalization:* We clarified [here](https://openreview.net/forum?id=Awsb8jhEx3&noteId=J7GWg2dRZb) that there are two main hyperparameters for $\texttt{DClAM}$, namely inverse temperature ($\beta$), and number of steps of AM ($T$). The number of AM steps ($T$) is usually 10-15. While $\texttt{DClAM}$ requires tuning $\beta$, it is a completely new strategy for deep clustering using associative memories that **outperforms** all the related baseline schemes. We clarified [here](https://openreview.net/forum?id=Awsb8jhEx3&noteId=zFRpWqtA2I) about the generalization of $\texttt{DClAM}$ to any network and dataset size.

- *evaluation metrics:* We clarified [here](https://openreview.net/forum?id=Awsb8jhEx3&noteId=q77ZFDaKHz)  and [here](https://openreview.net/forum?id=Awsb8jhEx3&noteId=ebazIWUcNv) that we also evaluated $\texttt{DClAM}$ on the supervised Normalized Mutual Information (NMI) metric where $\texttt{DClAM}$ **wins** for **5** out of the 8 datasets. We have also discussed why unsupervised metric like Silhouette Coefficient (SC) and reconstruction loss (RL) are necessary to evaluate unsupervised clustering methods and added new experiments on the Pareto frontier for all the hyperparameter configurations for different vision-based datasets to show how we selected best SC and RL across all datasets.

The responses were **acknowledged positively** by reviewer **[whZj](https://openreview.net/forum?id=Awsb8jhEx3&noteId=c0VelNjAMy)** and reviewer **[AgR2](https://openreview.net/forum?id=Awsb8jhEx3&noteId=dF0B4gr0fw)** and they have **increased** their score to **6**. We have not received any responses from reviewer ZsFx and Z3Ff while we have addressed all of their concerns.

**Revision Overview:**

- To enhance clarity, we have revised our introduction and related work sections, to outline the key differences between $\texttt{DClAM}$ and $\texttt{ClAM}$ and other deep clustering works.
- We have added the ablation study by varying the latent dimension $m$ for USPS in Appendix C.4, Fig 10. We have also clarified that the number of clusters $k$ is not a hyperparameter (end of sec A.2).
- We have added new experiments on the Pareto frontier for all the hyperparameter configurations for different vision-based datasets in Appendix C.2, and Figures 5,6,7,8,9.
- We have also included ablation results on the effect of varying the number of AM steps $T$ in Appendix C.4 and Figure 11, for both USPS and FMNIST datasets.

We are thankful for the valuable feedback from the reviewers, and we believe we have thoroughly addressed all the concerns raised by the reviewers in our responses and revised manuscript.

---

### Meta-Review · Area_Chair_BseA · 2024-12-20

**Metareview:**

This paper proposes a method for deep clustering based on an autoencoder model which applies the gradient descent step of an associative memory clustering method (ClAM) in the latent space of the encoder.  The reviewers are largely borderline, along with one more negative review.  Many reviewers note that the method appears to be largely similar to ClAM with the main distinction being incorporating the ClAM gradient descent operator in the latent space of an autoencoder.  The high level approach of incorporating a clustering metric on the latent space of an autoencoder is a well-known strategy in the deep clustering literature (though it also can often result in ill-posed formulations as well if not done carefully).  Here the authors argue that their method enjoys the advantage of not requiring a balancing between the sum of an autoencoder reconstruction loss and a clustering loss in the latent space as their method instead directly employs the gradient descent operator of ClAM in the latent space to drive latent points towards cluster centers.  However, while this does eliminate the need for balancing a weighting hyperparameters between two losses, one is still required to choose a hyperparameter for the number of time steps (T) used in applying the ClAM gradient descent steps.  The authors also provide an experimental evaluation of their approach which gives reasonable results, but some reviewers note that the datasets used are relatively small by modern standards.

Overall, the reviewers are largely lukewarm on the paper, and I am unfortunately inclined to agree with the critiques of the reviewers and recommend rejection.  Using an autoencoder with a clustering promoting loss or operator in the latent space is well established in the deep clustering literature.  To meet the bar for publication I would expect to see either a well motivated formulation along with strong theoretical analysis as to the merits of the formulation or convincing experiments on large scale datasets.  Here the authors have taken steps in both directions, but neither has led to the reviewers being particularly convinced about the approach.  I would encourage the authors to consider developing more extensive theoretical motivations for their proposed approach or evaluating the approach on larger scale datasets and look forward seeing an improved version of the work in future meetings.

**Additional Comments On Reviewer Discussion:**

The authors were largely responsive to the critiques of the reviewers in their rebuttal, leading to the reviewers who did respond to raise the scores, but not of the reviewers raised their scores to the point of arguing strongly for acceptance.

---

### Decision · Program_Chairs · 2025-01-22

Reject